

# Using hydraulic head, chloride and electrical conductivity data to distinguish between mountain-front and mountain-block recharge to basin aquifers

Etienne Bresciani[1,2], Roger. H. Cranswick[1,3], Eddie W. Banks[1], Jordi Batlle-Aguilar[1,4], Peter G. Cook[1],
Okke Batelaan[1]

[1]National Centre for Groundwater Research and Training, School of the Environment, Flinders University, Adelaide, SA 5001, Australia
[2]Korea Institute of Science and Technology, Seoul, 02792, Republic of Korea
[3]Department of Environment, Water and Natural Resources, Government of South Australia, Adelaide, SA 5000, Australia
[4]Kansas Geological Survey, University of Kansas, Lawrence, KS 66047, USA

*Correspondence to*: Etienne Bresciani (etienne.bresciani@flinders.edu.au)

**Abstract.** Numerous basin aquifers in arid and semi-arid regions of the world derive a significant portion of their recharge from adjacent mountains. Recharge can effectively occur through either stream infiltration in the mountain front zone (mountain-front recharge, MFR) or subsurface flow from the mountain (mountain-block recharge, MBR). While a thorough understanding of the recharge mechanisms is critical for water resource management, distinguishing between MFR and MBR is typically difficult. Here we present a relatively simple approach that uses hydraulic head, chloride and electrical conductivity data to distinguish between MFR and MBR. These types of data are inexpensive to measure, and in many cases are readily available from hydrogeological databases. In principle, hydraulic head can inform on groundwater flow directions and stream-aquifer interactions, while chloride can help to distinguish between different groundwater pathways if the sources have distinct concentrations. Electrical conductivity values can be converted to chloride concentrations using an empirical relationship, and hence can be used in a similar manner to chloride, thereby significantly increasing the data set. The practical feasibility and effectiveness of this approach are tested through the case study of the Adelaide Plains basin, South Australia, for which a wealth of historical groundwater level, chloride and electrical conductivity data is available. Hydraulic head data suggest that streams are gaining in the adjacent Mount Lofty Ranges and losing when entering the basin. They also indicate that not only the Quaternary sediments but also the underlying Tertiary sediments receive significant recharge from stream leakage in the mountain front zone. Chloride data also reveal clear spatial patterns suggesting that MFR dominates recharge of the low salinity groundwater found in the basin. This interpretation is further supported by stream water chloride analysis. This study demonstrates that both hydraulic head and chloride data can be effectively used to distinguish between MFR and MBR.



# 1 Introduction

Numerous basin aquifers in arid and semi-arid regions receive a significant portion of their recharge from adjacent mountains, as the latter typically benefit from higher rainfall and lower evapotranspiration (Wilson and Guan, 2004). Two recharge mechanisms can be recognized (Wahi et al., 2008): mountain-front recharge (MFR), which predominantly consists of stream infiltration in the mountain front zone, and mountain-block recharge (MBR), which consists of subsurface flow from the mountain towards the basin. Note that the mountain front zone is defined after Wilson and Guan (2004) as the zone of the basin located between the basin floor and the mountain block (Figure 1a). The term MFR has traditionally been used to encompass the two recharge mechanisms described above, but it may be more appropriate to use it for the first one only. The distinction between MFR and MBR is important, as each of these recharge mechanisms can imply quite different responses to land and water resource management practices as well as to climate change. A good understanding of these mechanisms is thus essential for an effective coordinated management approach of water resources in basins and adjacent mountains (Manning and Solomon, 2003; Wilson and Guan, 2004). Following Wahi et al. (2008), the collective process of MFR and MBR is referred to as mountain system recharge (MSR).

While various methods exist to estimate MSR as a bulk, characterizing MFR and MBR independently remains difficult. For instance, methods such as Darcy's law calculation and inverse groundwater flow modelling typically provide bulk MSR estimates (e.g. Hely et al., 1971; Anderson, 1972; Maurer and Berger, 1997; Siade et al., 2015). It is possible to consider MFR and MBR independently in a groundwater flow model, but the solution to the inverse problem is more likely to be non-unique (e.g. Bresciani et al., 2015b). The popular water balance and chloride mass balance methods also provide bulk MSR estimates when the measurements are made at the base of the mountain front zone or further downstream in the basin (e.g. Maxey and Eakin, 1949; Dettinger, 1989). Environmental tracers such as noble gases (e.g. Manning and Solomon, 2003), stable isotopes (e.g. Liu and Yamanaka, 2012) and radioactive isotopes (e.g. Plummer et al., 2004) can help to determine which of MFR or MBR is the dominant mechanism, but their analysis remains expensive and their interpretation can be difficult. The ultimate approach for characterizing MFR and MBR might be the integrated analysis of hydraulic, temperature and concentration data through the coupled modelling of groundwater flow, heat and solute transport in the combined basin-mountain system (e.g. Manning and Solomon, 2005) – but it is also arguably the most complex approach.

In this study, we explore alternatives to expensive and complex methods to investigate whether MSR to basin aquifers is dominated by MFR (Figure 1b) or MBR (Figure 1c), or if both types of recharge processes are significant (Figure 1d). We focus on the use of hydraulic head, chloride (Cl) and electrical conductivity (EC), which are inexpensive to measure and in many cases are readily available in large quantities in hydrogeological databases. The general utility of hydraulic head and Cl data to infer groundwater dynamics is well established (e.g. Domenico and Schwartz, 1997; Herczeg and Edmunds, 2000). Furthermore, EC values can be converted to Cl concentrations (as demonstrated later), and hence can be used in a similar manner to Cl. However, studies demonstrating the specific use of these data for the characterization of MSR mechanisms appear to be rare (Feth et al., 1966). This may reflect a traditionally low data density along mountain fronts,





which are not typically the prime locations for drilling groundwater wells due to the often complex hydrogeology and expected low aquifer yield (as these are the recharge areas). However, with an ever-growing number of wells accompanying development of basin and mountain areas, data density and spatial coverage steadily increases, even in these zones.

The Adelaide Plains basin in South Australia is used as a case study. This semi-arid region features a typical sedimentary basin bounded by a mountain range – the Mount Lofty Ranges, from which most of the recharge is believed to be ultimately derived (Miles, 1952; Shepherd, 1975; Gerges, 1999; Bresciani et al., 2015a). Groundwater in this basin has been used for about a century for industry, water supply and agriculture. Nonetheless, the relative contributions of MFR and MBR is still subject to debate, in part due to the enigmatic role of faults that run along the mountain front (Green et al., 2010; Bresciani et al., 2015a; Batlle-Aguilar et al., 2017).

## 2 Rationale

In this section, a generic rationale is presented for the use of both hydraulic head and Cl (or EC-derived Cl) data to distinguish between MFR and MBR to basin aquifers. Hydraulic head and Cl data can be used independently, but as they are of different nature, it is expected that their simultaneous use will result in a more complete and reliable characterization of the recharge mechanisms.

### 2.1 Using hydraulic head

Hydraulic heads directly relate to groundwater dynamics. Consequently, hydraulic head patterns could theoretically enable the identification of groundwater pathways, both in mountains and basins. Specifically, four types of analysis are suggested below that could inform the likely occurrence or absence of MFR and MBR:

1. Assessment of the correlation between hydraulic head and topography; in the mountain block, a good correlation would suggest that groundwater flow is dominated by local flow systems as opposed to regional flow systems (Tóth, 1963), implying that only a small portion of the recharge occurring over the mountain would make its way towards the basin. Therefore, MBR would be mostly limited to the recharge occurring over the so-called 'triangular facets' at the base of the mountain block, i.e. the in-between-streams zones in the lowest part of the mountain, over which recharge may be directly routed towards the basin through relatively shallow groundwater flow (Welch and Allen, 2012). In contrast, in the mountain front zone (i.e. in the upper basin zone), a good correlation between hydraulic head and topography would suggest that groundwater discharges to streams, so that MFR from stream leakage would be limited or non-existent.

2. Analysis of the shape of head contours adjacent to surface water features to identify losing and gaining stream conditions; it is well known that head contours show a curvature pointing in the downstream direction where the contour lines cross a losing stream (due to the mounding induced by groundwater recharge), whereas they show a curvature pointing in the upstream direction where the contour lines cross a gaining stream (due to the depression





induced by groundwater discharge) (e.g. Winter et al., 1998). Performing such analysis in the mountain block should indicate whether mountain groundwater appears mostly routed towards local streams, which would make it less likely for MBR to be significant. Additionally, performing such analysis in the mountain front zone should allow for testing the occurrence or absence of MFR (at least in the form of stream infiltration, which is the predominant form of MFR).

3.  Comparison of stream levels with nearby groundwater levels; a stream level higher than nearby groundwater levels would indicate a potential for groundwater discharge to stream, while the opposite would indicate a potential for stream infiltration (e.g. Winter et al., 1998). If data density is low, this analysis may be preferable over the previous one (#2) as it does not require head contours to be accurately determined. However, it can only inform on a *potential* interaction: groundwater discharge or recharge would be significant only if the hydraulic conductivity of the streambed is high enough. In contrast, the previous analysis (#2) could give a more definite answer because the curvature of head contours at some distance from the stream should only be visible if the groundwater-surface water interaction is significant relative to other flow components (i.e. horizontal flow).

4.  Evaluation of the vertical head gradient in the mountain front zone; recharge areas are associated with a decrease of hydraulic head with depth, while discharge areas are associated with an increase of hydraulic head with depth (e.g. Wang et al., 2015). Hence, in the mountain front zone, a head decrease with depth would suggest that MFR occurs (at a rate that depends on the vertical hydraulic conductivity), while in contrast an absence of head decrease (or a head increase) with depth would suggest that MFR does not occur.

In cases where faults run between the basin and the mountain, it may be tempting to study the difference of hydraulic head between the two sides of the fault zones, with the idea that a large difference would indicate that a fault zone constitutes a hydraulic barrier (e.g. Bense et al., 2013), and consequently that MBR would be low. However, a large difference in hydraulic head across a fault zone may not always imply that the fault zone constitutes a hydraulic barrier, as discussed in the following. Let us consider the hypothetical case of a sedimentary layer overlying a basement of relatively low hydraulic conductivity and which features a sharp transition in elevation as a consequence of faulting (Figure 2a). The hydraulic conductivity of the fault zone itself is assumed to be no different to that of the embedding materials. In this simple configuration, it happens that if the groundwater level below the fault (as a result of downstream controls) is lower than the elevation of the basement above of the fault, the groundwater level above the fault becomes essentially 'disconnected' from the lower system because it has to satisfy a minimum height (i.e. transmissivity) for groundwater to flow there (Figure 2b). Hence, in this case, a large difference in head can exist across the fault zone despite the fact that the fault zone itself has no specific (low) hydraulic conductivity. Furthermore, regardless of whether the fault zone has a low hydraulic conductivity or if it only implies a shift in basement elevation, the implications of a large difference in head in terms of the amount of flow eventually crossing the fault is far from obvious, as it depends on the hydraulic conductivity of either the fault or the basement – which is in either case difficult to determine. Besides, it can be argued that what is the most important is in fact to know whether or not the resulting hydraulic head above of the fault will be so high (relative to topography) as to imply





local groundwater discharge to mountain streams instead of deep lateral flow towards the basin. In other words, what matters is the partitioning of the mountain groundwater between these two pathways. This is precisely what the first three types of analysis presented above should contribute to determine.

## 2.2 Using chloride

Chloride (Cl) is a naturally occurring element in groundwater that is relatively non-reactive compared to other elements. This makes it a good conservative tracer of groundwater, except in particular cases where lithology can be an important source of Cl (e.g. Claassen and Halm, 1996). Thus, in many environments, the Cl concentration can be assumed to remain equal to that of recharge along the groundwater flowpaths (if the effects of dispersion can be neglected (e.g. Bresciani et al., 2014)). Therefore, if the Cl concentration of the potential MFR source has a distinct signature from that of the potential MBR source,

it could provide an excellent tool to distinguish between these two recharge mechanisms.

Cl in groundwater originates from atmospheric deposition, of which the rate depends on a number of factors including distance to the source (oceanic or terrestrial), elevation, terrain aspect, slope, vegetation cover and climatic conditions (Hutton and Leslie, 1958; Guan et al., 2010; Bresciani et al., 2014). Groundwater Cl concentrations also depend on evapotranspiration, which leaves Cl in solution, implying its enrichment (Eriksson and Khunakasem, 1969), and on the

spatial redistribution of recharge through groundwater flow. Hence, groundwater Cl concentration in mountains can be expected to show significant spatio-temporal variability. If MBR occurs, the associated Cl concentration depends on where (and when) the water initially originates – i.e. the infiltration point. On the other hand, if MFR occurs, the associated Cl concentration depends on streamflow generation mechanisms, i.e. overland flow, interflow and groundwater discharge. In particular, mountain streams are often supported in significant proportions by overland flow or interflow, in which case they

could have a lower Cl concentration than mountain groundwater since these mechanisms imply relatively little evaporation. Therefore, potential MFR water and potential MBR water are likely to have distinct Cl signatures.

In this study, the proposed strategy consists of analysing three types of water for Cl: groundwater in the basin, stream water at the mountain front, and groundwater in the mountain block near the basin. Assuming steady concentrations and conservative Cl, groundwater in the basin should have the same concentration as stream water at the mountain front if it

comes from MFR (assuming that transpiration from plants after stream infiltration and potential mixing with diffuse recharge are negligible). In contrast, the basin groundwater should have the same concentration as the mountain groundwater if it comes from MBR. For the latter to be assessed properly, it is important to assess groundwater Cl concentration in the mountain as close as possible to the basin to reduce potential risks of misinterpretation caused by the spatial variability of Cl concentration.

Electrical conductivity (EC) is known to be strongly correlated to Cl, and hence can be converted to Cl if a relationship between the two can be assumed. Ideally, an empirical relationship should be developed based on available pair measurements in the study area. As EC is typically more routinely measured than Cl, this should significantly increasing the data set.





## 3 Case study

### 3.1 Study area and background

The Adelaide Plains (AP) basin is a coastal sedimentary embayment of 1,700 km$^2$ in South Australia (Figure 3). The area is bounded by the Mount Lofty Ranges to the east and south, by the Light River to the north, and by the Gulf Saint Vincent to

the west. It can be split into two sub-basins: the Central Adelaide Plains (CAP) sub-basin south of Dry Creek, and the Northern Adelaide Plains (NAP) sub-basin north of Dry Creek. The topographic gradient is more pronounced in the CAP and adjacent mountains (regional slopes of about 0.8 % and 7 %, respectively) than in the NAP and adjacent mountains (regional slopes of about 0.3 % and 2.5 %, respectively). Torrens River and Gawler River are the main rivers in the CAP and in the NAP, respectively. A number of streams run down from the Mount Lofty Ranges, either feeding those rivers or

flowing directly into the ocean.

Rainfall is relatively low and potential evapotranspiration is high in this semi-arid area, with an average rainfall of 445 mm yr$^{-1}$ and an average maximum daily temperature of 21.6 °C at Adelaide Airport (station number 23034, 1970–2013; Australian Government, Bureau of Meteorology), which is located near the coast. Direct recharge from rainfall in the basin is thus expected to be relatively low, and instead most of the recharge is believed to be derived from the adjacent Mount

Lofty Ranges. The latter receive an average rainfall of 983 mm yr$^{-1}$ at Mount Lofty Cleland Conservation Park (station number 23810, 1970–2013; Australian Government, Bureau of Meteorology), i.e. more than twice that of the basin, and experiences cooler temperatures with an average maximum daily temperature of 15.2 °C at Mount Lofty (station number 23842, 1993–2007; Australian Government, Bureau of Meteorology).

The basin comprises complex sequences of Quaternary and Tertiary sedimentary deposits. The Quaternary sediments are

dominated by fluvio-lacustrine clay interbedded with sand and gravel, while the Tertiary sediments are dominated by sand, sandstone, limestone, chert, marl and shell remains interbedded with clay (Gerges, 1999). A number of faults dissect the basin, among which the Eden-Burnside Fault and the Para Fault are of primary interest in this study since they run along the foothill, almost at the margin of the CAP and the NAP sub-basins, respectively (Figure 3). The total thickness of the sedimentary units increases sharply downthrown of the major faults (up to 400 m in places). The thickness of the Quaternary

sediments ranges from 0 to almost 150 m across the basin (Figure 4a), while that of the Tertiary sediments ranges from 0 to about 500 m (Figure 4b). The Tertiary sediments are directly outcropping in the northeast part of the CAP. The basement of the basin and the Mount Lofty Ranges are mostly comprised of Proterozoic fractured rocks of various lithologies including slate, phyllite, quartzite, limestone and dolomite. Superficial sedimentary deposits also exist locally in the Mount Lofty Ranges.

Up to six semi-confined aquifers (named Q1 to Q6) are recognized in the Quaternary sediments from the central to western side of the basin (Gerges, 1999) (i.e. west of the mountain front zone). These aquifers contain water of variable salinity with a median value of around 1,300 mg L$^{-1}$. The underlying Tertiary sediments are generally subdivided into four aquifers (named T1 to T4) over a large part of the basin. However, there is no clear hydrogeological distinction between the various





Tertiary sediments along most of the mountain front zone in both sub-basins, and thus in this area they are considered to form a single undifferentiated Tertiary aquifer (Gerges, 1999; Zulfic et al., 2008; Baird, 2010). Salinity is relatively low in the upper aquifer (T1) with a median value of around 600 mg L$^{-1}$, and is higher in the deeper aquifers with median values of around 1,000 mg L$^{-1}$, 8,400 mg L$^{-1}$ and 40,000 mg L$^{-1}$ in T2, T3 and T4, respectively (note however that very few data are

available from the T3 and T4 aquifers). Because they present large areas of good salinity and yield, the T1 and T2 aquifers have been used since 1914 for occasional water supply, irrigation and industrial activities, and are currently the main targets of groundwater extraction in the AP (Gerges, 1999; Zulfic et al., 2008). Permanent, large cones of depression in both of these aquifers and forecasted increases in groundwater demand raise concerns about the sustainability of extraction in the coming years (Bresciani et al., 2015a). Risks are related to both potential depletion of the resource and rise in salinity, which

could make groundwater unusable. To better estimate these risks, a thorough understanding of the recharge mechanisms to these aquifers is necessary.

Early investigations suggested that the natural (i.e., pre-development) recharge to the Tertiary aquifers of the basin was dominated by stream infiltration along the mountain front (i.e., MFR) (Miles, 1952; Shepherd, 1975). In contrast, subsequent investigations suggested that the natural recharge of the Tertiary aquifers was dominated by subsurface flow from the Mount

Lofty Ranges (i.e., MBR) (Gerges, 1999, 2006). The latter conceptual model has formed the basis of most investigations of the Tertiary aquifers since its presentation, and underpinned the development of a number of groundwater flow and transport models of the basin aquifers (Jeuken, 2006a, b; Zulfic et al., 2008; Georgiou et al., 2011; Bresciani et al., 2015b). However, studies from Green et al. (2010) and Bresciani et al. (2015a) produced results supporting the hypothesis that both MFR and MBR could occur in significant proportions. To further investigate this question, the present study provides a re-appraisal of

available hydraulic head, Cl and EC data through application of the rationale described above.

## 3.2 Data sets

### 3.2.1 Hydraulic head data set

Hydraulic head data in the AP catchment (i.e. the area including both the basin and contributing mountain areas based on surface topography) were retrieved from the WaterConnect database (www.waterconnect.sa.gov.au, Government of South

Australia) on 04/11/2016. The collection dates span more than a century, the earliest measurements being from 1906 and the latest from 2016. The data were filtered out for unsuitable measurements such as measurements taken during pumping, aquifer test or drilling. After filtering, 111,538 hydraulic head measurements from 9,561 wells were obtained.

The data were subsequently split according to three aquifer groups: the AP Quaternary aquifers, the AP Tertiary aquifers ('AP' in these expressions will be omitted in the remaining text) and the Mount Lofty Ranges aquifers. Wells screened into

the basement of the basin were disregarded. This grouping is relevant in view of the hydrogeological characteristics of the system and the objective of the study. In particular, we did not distinguish between the T1 and T2 aquifers (i.e. the two main aquifers of the AP basin) because, as mentioned earlier, they are undifferentiated along most of the mountain front zone.



Furthermore, in the Mount Lofty Ranges, the presence of complex fracture networks and high relief can induce the blurring of otherwise depth-dependent signals, and so splitting the data according to depth may not be very meaningful, while it would also reduce data density.

The aquifer into which the wells were screened was informed in the database for about two thirds of the wells (6,209). For

the remaining wells, the aquifer group for the wells located in the basin was determined by comparing the well mid-screen elevation to the bottom elevation of the Quaternary sediments and to the top elevation of the basement (elevation surfaces by courtesy of the Department of Environment, Water and Natural Resources, South Australia). The largest number of wells was from the Quaternary aquifers (3,964), followed by the Mount Lofty Ranges aquifers (3,589) and the Tertiary aquifers (1,768).

Groundwater level fluctuations can be an issue for data interpretation. In particular, as this study focuses on natural recharge mechanisms, the impact of pumping constitutes a potentially important bias. It should be noted that the density of hydraulic head data is higher in areas of lower salinity groundwater, which coincides with areas that have experienced greater changes due to pumping. The measurements made before the main development period (i.e. before 1950) may have been less affected by pumping than more recent measurements, but limiting the analysis only to these measurements would dramatically reduce

the data density. In addition, even the earliest measurements may not be free of pumping influence, since it is likely that these were precisely taken to monitor the impact of pumping. Hence, instead of subjectively fixing an arbitrary date beyond which the data would be excluded, all available data were retained. For each of the wells that had multiple measurements, the temporal mean hydraulic head was calculated in an effort to smooth out the measurement errors and temporal fluctuations, and was be used in the analysis. The impact of pumping and natural fluctuations on the interpretation is

discussed later.

### 3.2.2 Chloride data set

Groundwater Cl data in the AP catchment were also retrieved from the WaterConnect database on 04/11/2016. The Cl data set was extended using the more commonly available EC data from the database. EC is known to be strongly correlated to Cl, and a relationship between EC and Cl was thus derived using 1,559 pair measurements (Figure 5). All EC data were

subsequently converted into Cl data using this robust relationship ($R^2 = 0.9996$). In total, 34,145 Cl or EC-converted Cl data (simply referred to as Cl data in the following) from 12,660 wells were obtained (i.e. slightly more than for hydraulic heads due to a less restrictive filtering). The collection dates span the same period as for the hydraulic head data.

The same three aquifer groups were distinguished as for the hydraulic head data, and the same procedure was also applied to determine the aquifer group into which the wells are screened. The largest number of wells was from the Quaternary aquifers

(4,963), followed by the Mount Lofty Ranges aquifers (4,395) and the Tertiary aquifers (2,963).

Pumping may also have impacted Cl concentrations. Namely, as for hydraulic head data, the density of Cl data is higher in areas that have experienced pumping. However, the impact of pumping on Cl concentrations is expected to be less important than on hydraulic heads because groundwater chemistry typically responds less rapidly to perturbations than groundwater





hydraulics. Hence, as for hydraulic heads, all available Cl data were retained. For each of the wells that had multiple measurements, the temporal mean Cl concentration was calculated and was used in the analysis.

Flow rate and EC data from a number of streams running down from the Mount Lofty Ranges into the AP basin were also retrieved from the WaterConnect database. Six gauging stations were located close enough to the mountain front zone to be
relevant to the current study. Details on this data set are given in Table 1. The reported EC values of surface water were converted into Cl concentrations using the same relationship as developed for groundwater, which is deemed appropriate given the common origin of these waters.

### 3.3 Data analysis

#### 3.3.1 Hydraulic heads

A hydraulic head map was constructed for each of the three aquifer groups (Quaternary aquifers, Tertiary aquifers and Mount Lofty Ranges aquifers) (Figure 6). The choice of the interpolation method and parameters used to construct these maps was critical. The Inverse Distance Weighting method would produce the famous 'bull's eye' effect around single data points, which would compromise the interpretation of head contours. Instead, the Diffusion Kernel interpolation method from the Geostatistical Analyst extension of ArcGIS 10.4.1 was used. This method allows for a more realistic interpolation
when the underlying phenomenon governing the data is diffusive, as is the case for hydraulic head. The most important parameter in this method is the bandwidth, which is used to specify the maximum distance at which data points are used for prediction. Taking this parameter too small would undermine the prediction capability (i.e. many areas will remain uncovered by the interpolation), while taking it too large would produce overly smoothed results. This parameter was set to 1,200 m in all three cases. This value allowed for a relatively good interpolation coverage while retaining most relevant
spatial head variations.

Figure 6a displays head contours in the Quaternary aquifers and the Mount Lofty Ranges aquifers, while Figure 6b displays head contours in the Tertiary aquifers and the Mount Lofty Ranges aquifers. Only the most relevant area of the catchment is shown for a better visualization. The colours and contour interval are different in the basin and in the Mount Lofty Ranges to accommodate the fact that the range of head variations is much larger in the mountain than in the basin. Topographic
contours are shown with the same interval as for hydraulic head contours (i.e. different in the basin and the mountain). The topographic contours were calculated after application of a circular moving-average window of 1,200 m radius to the topographic map (i.e. matching the bandwidth used in the interpolation method for hydraulic head) to facilitate comparison with the hydraulic head contours. The figures reveal that the shape of hydraulic head contours and topographic contours is quite similar in the Mount Lofty Ranges aquifers, indicating a good correlation between hydraulic head and topography. This
suggests, at least qualitatively, that groundwater flow is dominated by local flow systems in the Mount Lofty Ranges, and by consequence that the source of MBR may be limited to the recharge occurring over 'triangular facets' at the base of the mountain (see section 2.1). In contrast, head contours do not appear to follow a subdued expression of topographic contours





in the mountain front zone, both in the Quaternary and Tertiary aquifers. This suggests that the streams are at least not gaining, and thus potentially losing.

Figure 7a and Figure 7b are essentially the same as Figure 6a and Figure 6b, respectively, but rivers are shown and not topographic contours. Different sets of figures appeared necessary to improve the readability and allow for a more focused interpretation. The shape of head contours near streams is generally indicative of gaining conditions in the Mount Lofty Ranges. Exceptions are principally located along the upper reaches of rivers, i.e. where the stream order is small. The latter observation suggests that streams are not primarily initiated by groundwater discharge but by overland flow or interflow, and, as a consequence, that the infiltration capacity of the mountain block is limited. In contrast, in the mountain front the shape of head contours near streams is in most instances indicative of losing conditions. A striking symmetry is even observed for some of the main rivers entering the basin, with head contours pointing upstream at the base of the Mount Lofty Ranges while pointing downstream in the mountain front zone, indicating a sudden change of conditions from gaining to losing (e.g. along the Gawler River and its tributaries). Remarkably, indications of losing river conditions are observed not only in the Quaternary aquifers but also in the Tertiary aquifers, suggesting that significant amounts of water losses to the Quaternary aquifers reach the underlying Tertiary aquifers.

Figure 8a and Figure 8b display the result of the subtraction, at every point, of the nearest river elevation by the hydraulic head. The first figure shows the result for the Quaternary aquifers and the Mount Lofty Ranges aquifers, while the second figure shows the result for the Tertiary aquifers and the Mount Lofty Ranges aquifers. In the Mount Lofty Ranges, these figures corroborate the interpretations made above regarding the groundwater-surface water interactions: most rivers appear to be potentially gaining (as seen from the blueish-coloured areas, which indicate a potential for groundwater to flow towards the nearest river), except in their upper reaches (as seen from the reddish-coloured areas, which indicate a potential for groundwater to receive water from the nearest river). The Quaternary aquifers are revealed as potentially receiving water from rivers over the entire basin, and especially in the mountain front zone, where the difference between nearest river elevation and hydraulic head is the largest. The Tertiary aquifers globally show the same patterns, except over a few small areas near the mountain front in the CAP sub-basin, where a potential for groundwater to flow towards streams is indicated – namely around a portion of Torrens River. The fact that no area shows up as potentially gaining in the western part (i.e. the lower part) of the basin can be surprising, as one may expect to find groundwater discharge areas here, particularly near the coast. Under pre-development conditions, the hydraulic head in these areas was indeed higher than the land surface (Gerges, 1999). There is no doubt that this observation reveals the effect of pumping, which is known to be especially intense in the western part of the basin in both the T1 and T2 aquifers (e.g. Bresciani et al., 2015a).

The vertical head gradient in the basin was investigated through the head difference between the Quaternary and Tertiary aquifers. The results show that the most of the mountain front zone is characterized by a significant downward head gradient, with up to 59 m head difference (Figure 9). This indicates a downward leakage of groundwater from the Quaternary to the Tertiary aquifers. The rate at which this leakage occurs is of course also function of the effective vertical hydraulic conductivity and relevant distance between these units, which are largely unknown. Note that in Figure 9 the large red zone





located towards the centre of the NAP near the Gawler River reflects the impact of extensive historical and ongoing groundwater extraction from the T2 aquifer.

### 3.3.2 Chloride concentrations

A Cl concentration map was constructed for each of the three aquifer groups (Quaternary aquifers, Tertiary aquifers and
Mount Lofty Ranges aquifers) using the Inverse Distance Weighting interpolation method from the Geostatistical Analyst extension of ArcGIS 10.4.1. This method is appropriate for the Cl values because the focus is not on the contours, and hence the 'bull's eye' effect is not really an issue. Furthermore, Cl does not result from a diffusive process at regional scale (advection typically dominates at this scale), and so the Diffusive Kernel method would be inappropriate. The Inverse Distance Weighting interpolation method also has the advantage of being exact at the data points. The power parameter was
set to 2 and a standard neighbourhood was used with 15 maximum neighbours and 10 minimum neighbours.

Figure 10a shows the Cl concentrations in the Quaternary aquifers and the Mount Lofty Ranges aquifers, while Figure 10b shows the Cl concentrations in the Tertiary aquifers and the Mount Lofty Ranges aquifers. The figures reveal a strong correlation between stream locations and low Cl concentration zones in the basin, both in the Quaternary aquifers and Tertiary aquifers. It seems highly unlikely that such a correlation would be observed if MBR was the main recharge
mechanism. Furthermore, no such correlation can be seen in the Mount Lofty Ranges aquifers. Here, the Cl concentration appears correlated with elevation, with lower values occurring at higher elevations. This trend is expected, since the rate of evapotranspiration – which largely controls Cl concentration – is expected to decrease with elevation as a result of higher rainfall and lower temperature. In line with these observations, there is a clear discontinuity in Cl concentration at the transition between the mountain and the basin, almost everywhere along the front line. This suggests that little or no
hydraulic connection occur between the mountain and the basin through the subsurface. In particular, the lowest concentrations found along streams in the basin aquifers are in most cases lower than the concentrations observed just at the base of the mountain, suggesting that this water originates from stream leakage in the basin (i.e. MFR). The possibility that this water originates from the higher elevation areas of the mountain – where salinity is low – through deep groundwater flowpaths is unlikely since the hydraulic head data analysis suggests a predominance of local flow systems in the Mount
Lofty Ranges (section 3.2.1), and since groundwater salinity in the deep layers of the basin generally show high salinity. Furthermore, the Cl concentrations in the in-between streams zones (i.e. away from streams) in the basin aquifers are in most cases much higher than in the 'triangular facets' of the base of the mountain, and so this suggests that these 'triangular facets' do not contribute either to the basin recharge (or at least not in significant proportion).

The Cl concentration in streams running down from the Mount Lofty Ranges into the AP basin was also investigated to see
if stream leakage can explain the observed groundwater concentrations in the basin. A summary of available flow rate, electrical conductivity and derived Cl concentration data for six monitoring stations located at the transition between the mountain and the basin is presented in Table 1. The location of the stream gauges is indicated in Figure 3. The relationship between flow and Cl is shown from a scatter plot in Figure 11. The stream Cl concentration displays significant temporal



variations with a clear decreasing trend as flow increases. The relationship between flow and Cl concentration varies between different streams, reflecting different catchment characteristics (i.e. topography, climate, geology, landuse) which are likely to influence the streamflow generation mechanisms. Time series are presented for the selected cases of Gawler River and Brownhill Creek in Figure 12a and Figure 12b, respectively. The time series confirm that periods of low stream Cl
concentration consistently coincide with periods of high flow, during which low Cl concentrations can be explained by a relatively large contribution of overland flow or interflow which should experience little evapotranspiration relative to groundwater discharge (the significance of overland flow or interflow to streamflow generation is also supported by the hydraulic head data analysis, see section 3.2.1). During high flow periods, the infiltration potential in the mountain front zone should be enhanced due to higher stream water levels and larger wetted areas combined with losing conditions. Hence,
in the absence of more quantitative constraints on the timing of stream leakage, we propose to take the flow-weighted average Cl concentration in streams as a representative value for MFR Cl concentration. The flow-weighted average Cl concentration is 221, 115, 146, 67, 107 and 91 mg L$^{-1}$ in the North Para River, South Para River, Gawler River, Dry Creek, First Creek and Brownhill Creek, respectively (Table 1). These data show that streams are a plausible source for the low Cl concentrations observed in the basin aquifers (see Figure 10).

**4 Discussion**

**4.1 Strengths and limitations of using hydraulic head and chloride data**

One of the main strengths of hydraulic head data is that they can unambiguously indicate the contemporary flow direction (if hydraulic conductivity is considered to be isotropic). In this study the analysis of head contours gave indications that groundwater flows for a large part in local systems feeding streams in the Mount Lofty Ranges. It also allowed for the
identification of losing stream conditions in the mountain front zone, where leakage from streams appears to recharge not only the Quaternary aquifers but also the Tertiary aquifers of the AP basin in significant proportions. Studying the head variation with depth in the basin gave further evidence that downward groundwater flow from the Quaternary aquifers to the Tertiary aquifers is occurring. The rate of this flow is however unknown as long as the effective vertical hydraulic conductivity is unknown.
The main limitation of hydraulic head data is probably that they are quite sensitive to pumping, as illustrated through the present case study. This is problematic when the objective is to study the natural (i.e. pre-development) recharge mechanisms. Pumping in the AP basin mostly affects groundwater levels in the western part of basin, where large cones of depression exist in the Tertiary aquifers due to extensive historical and ongoing pumping. Therefore, for the purpose of this study which focuses on the eastern part of the basin (where the mountain front zone is located), the issue may not be as
critical. However, smaller-scale pumping wells are likely to exist in the mountain front zone as well and may affect the results to an unknown degree. The fact that the degree to which the hydraulic head data may have been distorted by pumping is unknown represents a source of uncertainty likely to be non-negligible.



Cl is a potentially great tool to distinguish between MFR and MBR, but its usefulness depends on how different the sources are in terms of Cl concentration. That is, if all the potential sources of recharge had the same Cl concentration, nothing could be learnt about recharge and flow mechanisms on the basis of Cl data. Fortunately, different processes involved in the generation of MFR and MBR imply that these two potential sources of water are likely to have different Cl signatures (see

section 2.2). This is certainly the case in the present case study, where the Cl concentration at the base of the mountain (i.e. potential MBR source) is seen to be significantly different from that of the basin, while in contrast the Cl concentration in rivers (i.e. potential MFR source) is seen to be similar to that of the low-concentration zones of the basin, which are located adjacent to the surface water features. These observations allowed for a non-ambiguous interpretation of the recharge mechanisms (i.e. MFR appears as a dominant mechanism in the AP basin).

As for hydraulic heads, pumping can potentially distort the Cl concentrations from those of the undisturbed system. However, solute concentrations are expected to be less sensitive to pumping than hydraulic heads, namely because short travel times in groundwater imply that a dramatic shift in Cl concentrations is unlikely to be seen in Cl concentrations away from the main pumping centres. Furthermore, if recharge from streams in the basin was only induced as a result of recent pumping, groundwater should have a very modern (post-development) recharge signature. In most of the AP basin,

groundwater dating shows that this is not the case (Batlle-Aguilar et al., 2017). It can also be noted that correlation between low salinity zones and streams was already observed in the 1950s, i.e. using measurements anterior to the main groundwater development period (Miles, 1952).

The interpretation of Cl data also relies on the assumptions of constant Cl inputs. This assumption may not be strictly satisfied over the entire AP basin because groundwater in the Tertiary aquifers can be quite old, as revealed by numerous

samples showing paleo-meteoric origin (> 12,000 y) according to carbon-14 dating and noble gas measurements (Batlle-Aguilar et al., 2017). This indicates that different climatic conditions might have prevailed at the time of recharge, implying possible variations in Cl inputs. However, such old groundwater is mostly observed in the western part of the basin; groundwater in the mountain front zone is much younger (in most cases between modern and < 10,000 y according to carbon-14 dating), making it less likely for these to reflect drastically different climatic conditions. Furthermore, even if the

Cl inputs did vary over time, such temporal variations would not in itself explain the correlation of groundwater Cl concentration with streams in the basin.

Finally, the assumption of conservative Cl is deemed reasonable because chloride is usually not strongly adsorbed to mineral surfaces, and the aquifer materials in the study area are not expected to be a significant source of Cl in comparison to atmospheric inputs. In addition, even if Cl was not strictly conservative, leakage from streams would again appear necessary

to explain the observed correlation of groundwater Cl concentration with streams in the basin.

## 4.2 MFR versus MBR in the AP basin

In an early study, Miles (1952) noted that the pre-development groundwater levels along the mountain front of the AP basin were reflective of unconfined conditions, and that the subsurface materials in this zone were favourable to stream infiltration.



In addition, Miles (1952) already analysed the groundwater salinity distribution and observed that salinity contours were forming fan-shaped zones of low salinity 'mushrooming' outwards from streams, with such patterns being visible up to more than 100 m below the ground surface. He concluded that stream infiltration along the mountain front zone was a major recharge mechanism for the basin aquifers. Later, in a study of the NAP aquifers, Shepherd (1975) arrived to the same

conclusion partly using similar arguments and further noting that: (i) groundwater hydrographs in the Quaternary aquifers were each year showing a rapid rise in water level shortly after Gawler River and Little Para River started to flow; (ii) the vertical head gradient and vertical hydraulic conductivity were indicative of significant downward flow from the Quaternary to the Tertiary aquifers. Additionally, a number of studies directly measured groundwater gains and losses using differential flow-gauging along streams entering the AP basin (Hutton, 1977; Green et al., 2010; Cranswick and Cook, 2015), and all

found that several streams were losing a significant amount of water in the mountain front zone. Finally, Zulfic et al. (2010) found that airlift yields (a proxy for transmissivity) in the Mount Lofty Ranges did not increase beyond 100 m depth for most geology types. This can be interpreted as hydraulic conductivity being relatively low beyond that depth, thus promoting local groundwater flow systems in the mountain, in line with the current analysis.

In contrast, Gerges (1999) and Batlle-Aguilar et al. (2017) proposed that MBR is the dominant recharge mechanism for the

Tertiary aquifers of the AP basin. A major argument used in these studies was based on the observation that salinity is generally higher in the Quaternary aquifers than in the T1 and T2 aquifers. From this observation, the authors suggested that the water found in the T1 and T2 aquifers could not be the result of downward leakage, and that instead it had to come from the Mount Lofty Ranges through subsurface flow. However, along streams, the Cl concentration in the Quaternary aquifers is in fact very similar to that of the underlying Tertiary aquifers across a large eastern part of the basin (Figure 10). This

makes the above reasoning invalid. Another argument used by Batlle-Aguilar et al. (2017) was based on the observation that relatively old groundwater was measured near the top of Tertiary aquifers (from carbon-14 dating). From this observation, the authors suggested that groundwater could not be recharged in the basin, but rather further away, in the Mount Lofty Ranges. However, most of the old groundwater was only found quite some distance away from the mountain front zone, where the Tertiary aquifers become confined (logically implying an increase of age with distance from the recharge zone).

Furthermore, in contrast relatively young groundwater was found at significant depth near major faults of the basin, precisely suggesting the occurrence of focused recharge (Batlle-Aguilar et al., 2017). Finally, neither Gerges (1999) nor Batlle-Aguilar et al. (2017) proposed a mechanism to explain how the groundwater could be more saline outside of the low-salinity corridors, as yet consistently observed across the basin, not only in the Quaternary aquifers but also in the Tertiary aquifers (Figure 10). These zones of higher salinity directly contradict the hypothesis that the aquifers would get recharged from

subsurface flow of low-salinity groundwater occurring in the Mount Lofty Ranges. The more saline groundwater found in these zones more likely results from the percolation of diffuse recharge in the basin, as this water would experience significantly higher evapotranspiration than the water found in streams running down from the mountain, thus yielding higher solute concentrations.





Hence, on the basis of the robust evidences given in this work and through a critical review of earlier investigations, this study proposes that infiltration from streams into the Quaternary aquifers and subsequent downward migration towards the Tertiary aquifers is the most plausible and predominant recharge mechanism for the low-salinity groundwater found in the aquifers of the AP basin (i.e. as in Figure 1b). This finding is expected to have important consequences for future investigations and the management of water resources in the region of Adelaide. A conceptual model depicting the suggested recharge mechanisms, and how they can explain the observed Cl (or salinity) patterns in the eastern part of the basin, is shown in Figure 13.

## 5 Conclusion

This study presented and demonstrated through an example the effectiveness of using hydraulic head, Cl and EC data to distinguish between MFR and MBR to basin aquifers. The most useful way of using hydraulic head data is through the analysis of the shape of head contours adjacent to surface water features to identify losing and gaining stream conditions in the mountain front zone as well as in the mountain itself. Other useful ways of using hydraulic head data were presented: analysis of the degree of correlation of hydraulic head with topography, comparison of stream levels with nearby groundwater levels, and evaluation of the vertical head gradient in the mountain front zone. However, the latter three methods can only indicate the flow direction, while the significance of this flow relative to other flow components (i.e. horizontal flow) remain unknown as long as hydraulic conductivity is unknown. In contrast, the former method should only reveal losing or gaining conditions if the associated flow rates are significant relative to other flow components.

Groundwater level fluctuations and especially the effects of pumping can be an issue for the interpretation of hydraulic head data when the objective is to study natural recharge mechanisms. The interpretation of Cl (or EC) data is not free of such issues, but these data are likely less sensitive to perturbations. In addition, the case study demonstrated that very clear spatial patterns in the Cl distribution can be observed in the field that leave little room for ambiguity in the interpretation, including when considering the fact that some of the underlying assumptions may not be entirely satisfied.

Compared to methods that use noble gas, radioactive or isotopic tracers, the proposed approach appears simpler, more cost effective, and more reliable due to the much higher data density generally achievable (i.e. given current technologies and budget constraints). Nevertheless, in contrast with some of these methods (e.g. noble gases), this approach is only qualitative, i.e. it does not allow for the quantification of the relative proportion of MFR and MBR, neither than for their absolute amount. One way to extend the ideas presented in this study to gain more quantitative insight would be to use the data as calibration targets in a groundwater flow and Cl transport model. The degree to which the recharge rates could be constrained through this approach is the subject of ongoing research (Bresciani et al., 2015b).





**Data availability**

Groundwater hydraulic head, Cl and EC data used in this study are available on the WaterConnect database (https://www.waterconnect.sa.gov.au).

**Acknowledgements**

This study was supported by the Goyder Institute for Water Research through the project I.1.6 "Assessment of Adelaide Plains Groundwater Resources" (2013–2015). It was also supported by the Korea Research Fellowship Program through the National Research Foundation of Korea (NRF) funded by the Ministry of Science, ICT and Future Planning (project number 2016H1D3A1908042).

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





| | North Para River (A5050502) | South Para River (A5051009) | Gawler River (A5050505) | Dry Creek (A5041051) | First Creek (A5040578) | Brownhill Creek (A5040901) |
|---|---|---|---|---|---|---|
| Mean EC (uS/cm) | 3424 | 1414 | 3121 | 1579 | 1276 | 808 |
| **Flow-weighted mean EC (uS/cm)** | **1095** | **656** | **758** | **411** | **611** | **545** |
| 10th percentile Cl (mg/L) | 214 | 99 | 183 | 37 | 76 | 87 |
| 90th percentile Cl (mg/L) | 852 | 304 | 1615 | 184 | 259 | 191 |
| Median Cl (mg/L) | 1441 | 457 | 600 | 990 | 432 | 144 |
| Mean Cl (mg/L) | 846 | 288 | 769 | 353 | 257 | 145 |
| Mean flow rate (GL/y) | 10.25 | 4.98 | 30.07 | 6.28 | 2.28 | 1.92 |
| **Flow-weighted mean Cl (mg/L)** | **221** | **115** | **146** | **67** | **107** | **91** |
| # EC record days | 6064 | 1221 | 131 | 944 | 937 | 29 |
| Records period | 1994–2016 | 2003–2010 | 1970–1995 | 2013–2016 | 2013–2016 | 2012–2016 |

**Table 1. Surface water flow rate, EC values and their conversion into Cl concentrations for the six gauging stations located near the mountain front zone (gauging station number in parenthesis; see Figure 3 for site locations). Note that the North Para River and South Para River join about 1 km downstream of the Para Fault to form the Gawler River.**



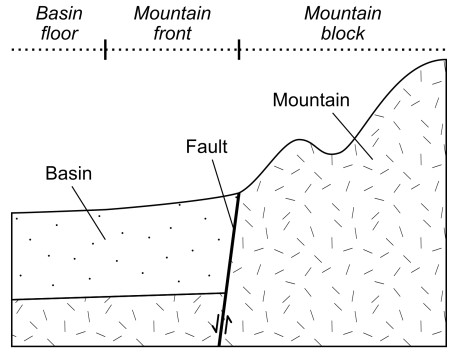

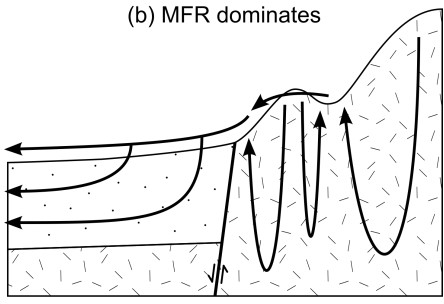

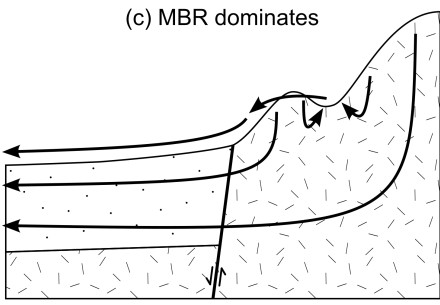

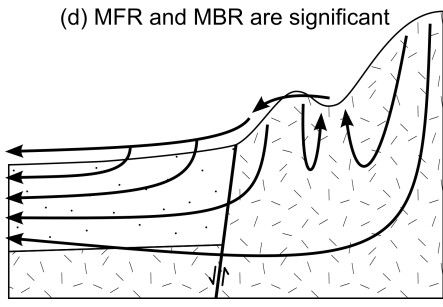

**Figure 1. Conceptual models of the transition between mountain and basin. (a) Physical configuration. (b-c) Three possible conceptualizations regarding the MSR to the basin: (b) MFR dominates, (c) MBR dominates, and (d) both MFR and MBR are significant.**





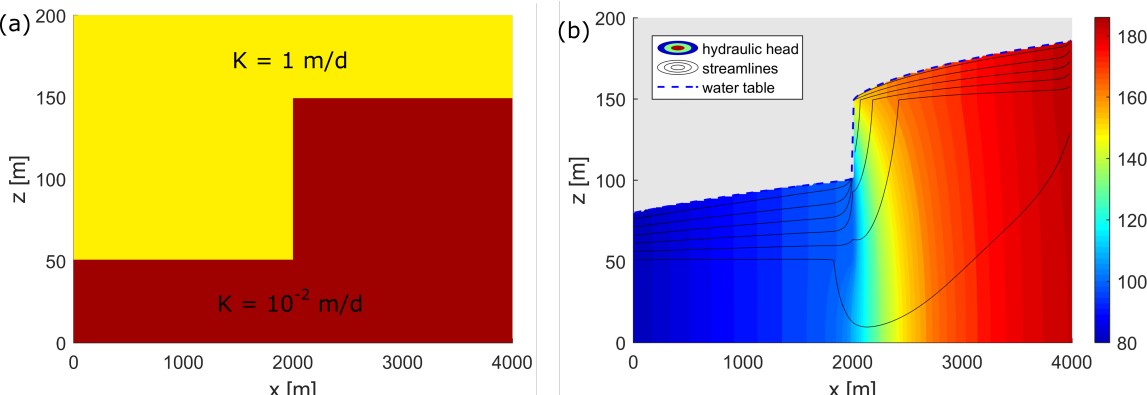

**Figure 2. Impact of a difference in basement elevation induced as a result of faulting on hydraulic head in a hypothetical setting. (a) Hydraulic conductivity field in a vertical cross-section representing a sedimentary layer (in yellow) overlying a basement having significantly lower hydraulic conductivity (in brown), with the fault zone material itself having no different hydraulic conductivity (i.e. the fault only implies the difference in basement elevation). (b) Results from an unconfined groundwater flow simulation in which a constant head (80 m) was specified on the left boundary and inflow was specified on the right boundary (at a rate proportional to the hydraulic conductivity). A sharp difference in hydraulic head is observed across the fault zone. The simulation was performed using MODFLOW-NWT (Niswonger et al., 2011) with uniform grid spacing (200 cells in horizontal direction × 150 cells in the vertical direction).**







**Figure 3. AP basin and AP catchment based on surface topography. Elevations are in mAHD (Australian Height Datum, i.e. above mean sea level).**





**Figure 4. Thickness of the Quaternary (Q) sediments (a) and Tertiary (T) sediments (b) in the AP basin. The colour scheme is different for the two figures. The maps were constructed by calculating the difference between the land surface elevation and the bottom elevation of the Quaternary sediments (a), and between the bottom elevation of the Quaternary sediments and the top elevation of the basement (b) (elevation surfaces by courtesy of the Department of Environment, Water and Natural Resources, South Australia).**





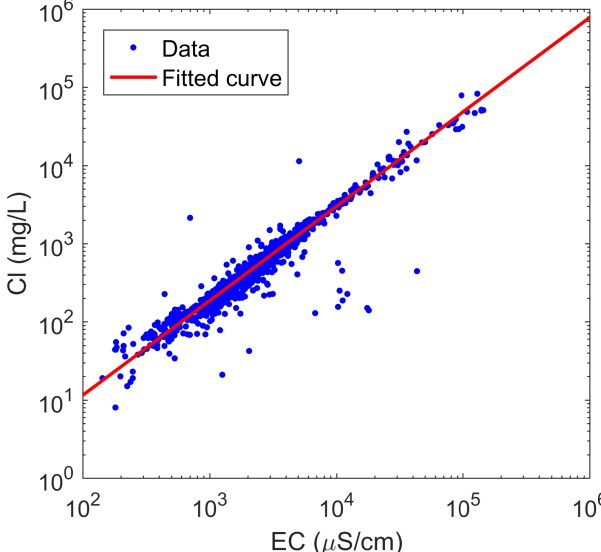

**Figure 5. Cl versus EC data in the AP catchment. The fitted function used to describe the relationship is $[Cl] = 0.04411 \times [EC]^{1.208}$, where [Cl] and [EC] are in units of mg L$^{-1}$ and µS cm$^{-1}$, respectively (coefficient of determination: $R^2 = 0.9996$).**







**Figure 6.** Head contours in the Quaternary aquifers and Mount Lofty Ranges aquifers (a), and in the Tertiary aquifers and Mount Lofty Ranges aquifers (b). Greenish colours are associated with head values in the basin, while brownish colours are associated with head values in the mountain. Topographic contours are also indicated. The contour interval is different in the basin (10 m) and in the Mount Lofty Ranges (40 m) both for head and topographic contours to accommodate the fact that the range of 5 variations is much larger in the mountain than in the basin. Values are in mAHD (Australian Height Datum).





**Figure 7. Head contours in the Quaternary aquifers and Mount Lofty Ranges aquifers (a), and in the Tertiary aquifers and Mount Lofty Ranges aquifers (b). Greenish colours are associated with head values in the basin, while brownish colours are associated with head values in the mountain. The head contour interval is different in the basin (10 m) and in the Mount Lofty Ranges (40 m) to accommodate the fact that the range of variations is much larger in the mountain than in the basin. Values are in mAHD (Australian Height Datum).**





**Figure 8. Result of the subtraction, at every point, of the nearest river elevation by the hydraulic head, in the Quaternary aquifers and Mount Lofty Ranges aquifers (a), and in the Tertiary aquifers and Mount Lofty Ranges aquifers (b). The same colour scheme is applied everywhere. Blueish colours are for negative values, indicating that the nearest river level is lower than the hydraulic head, while reddish colours are for positive values, indicating that the nearest river level is higher than the hydraulic head.**





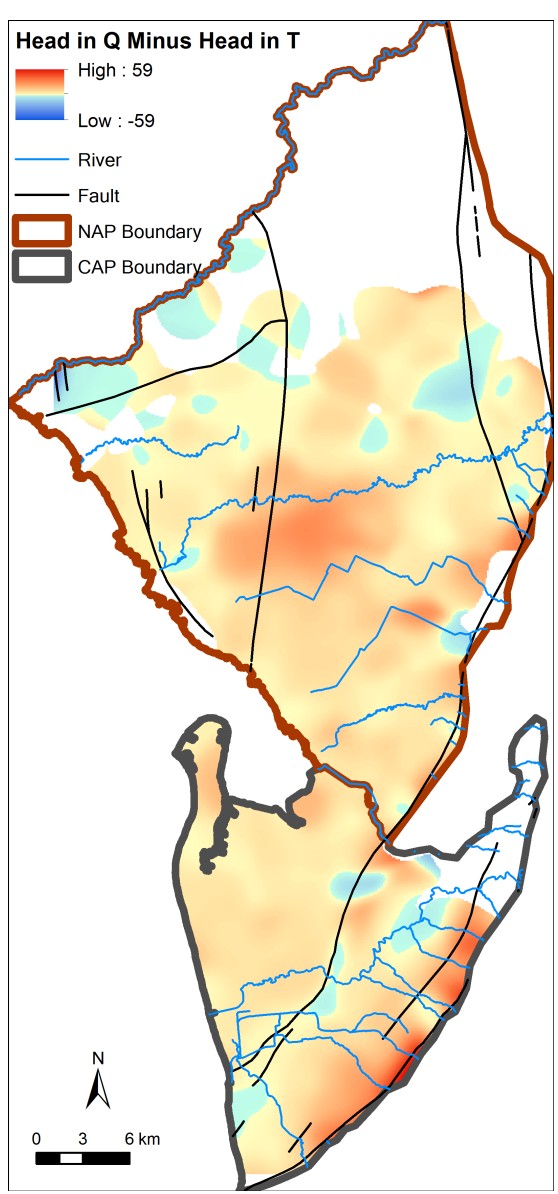

**Figure 9. Head difference between the Quaternary aquifers (Q) and the Tertiary aquifers (T). Blueish colours are for negative values, indicating that the head is lower in Q than in T, while reddish colours are for positive values, indicating that the head is higher in Q than in T.**





**Figure 10. Chloride concentration in the Quaternary aquifers and Mount Lofty Ranges aquifers (a), and in the Tertiary aquifers and Mount Lofty Ranges aquifers (b). The colour scheme is chosen favourable to the study of relatively low salinity zones, i.e. [Cl] < 1,400 mg L$^{-1}$ (or [EC] < 5,327 μS cm$^{-1}$) for the purpose of this study; much higher values exist and are all represented in red.**




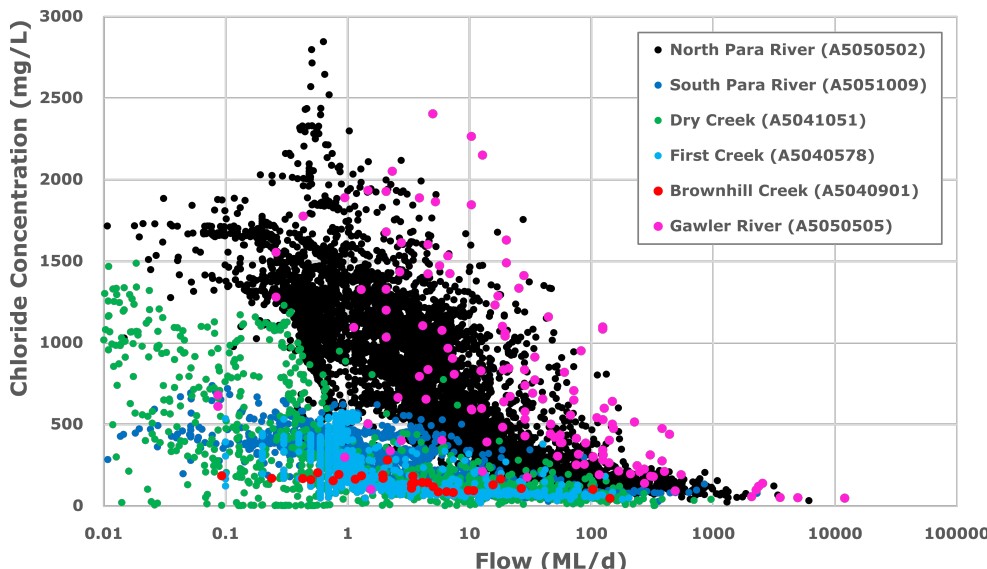

**Figure 11. Scatter plot of chloride concentration versus stream flow for six streams running down from the Mount Lofty Ranges into the AP basin.**



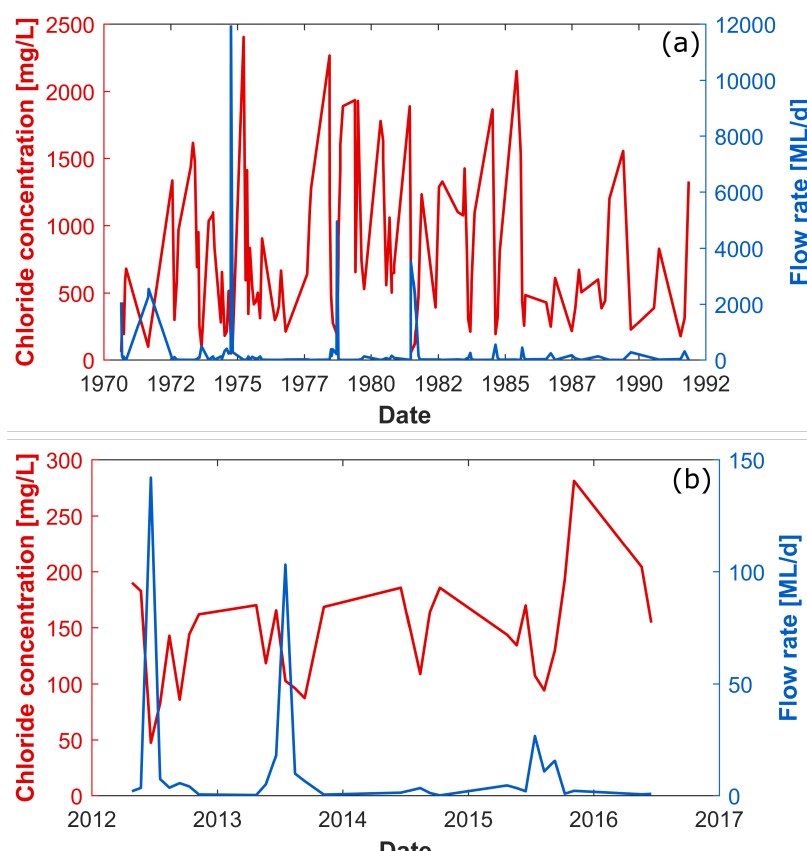

**Figure 12. Surface water chloride and flow data for Gawler River (a) and Brownhill Creek (b). Notice the different scales on the axes.**




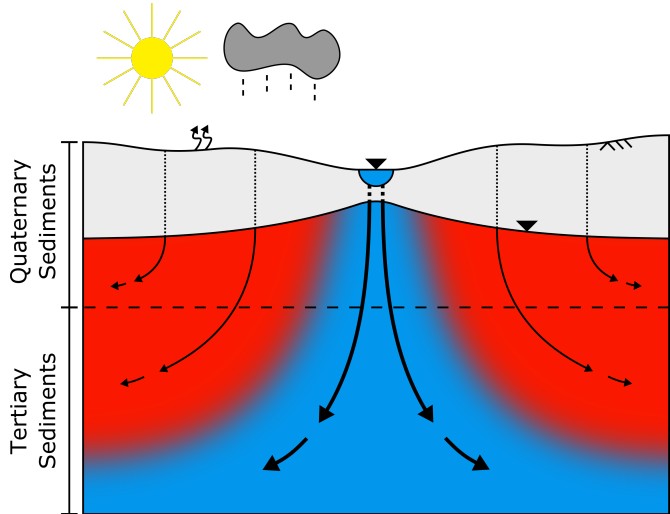

**Figure 13. Conceptual model of the recharge mechanisms for the AP basin aquifers, as seen in a cross-section perpendicular to (and centred on) a stream in the mountain front zone, and how they can explain the observed salinity patterns in the eastern part of the basin. In blue: groundwater having relatively low salinity; in red: groundwater having relatively high salinity.**