# Peer review of "Using hydraulic head, chloride and electrical conductivity data to distinguish between mountain-front and mountain-block recharge to basin aquifers"

_Hydrology and Earth System Sciences, 2017_

## Referee Comment (RC1) · Anonymous Referee #1 · 18 Sep 2017

General Comments: This manuscript presents research that was completed to determine the source(s) of recharge to the Adelaide Plains, Australia. The authors were specifically interested in determining whether recharge in the Adelaide Plains was sourced from mountain-block recharge (MBR) or mountain-front recharge (MFR) or a combination of the two in the nearby Mount Lofty Range using a combination of regional hydraulic head data and chloride data. The article is well written and will likely gain broad readership within the hydrogeology and mountain hydrology communities. It has immediate regional implications to the evolving conceptual framework for the

Adelaide Plains. There are a couple of weaknesses in its current form that prevent a recommendation for acceptance.

First, the Abstract, Introduction, and Conclusions do not do the article justice. The Abstract needs data in my opinion, but this is admittedly difficult since most of the results are presented spatially in maps. In any case, the Abstract feels methods-heavy since only the last 6 sentences discuss actual results. This could be remedied with a concise problem statement in the Abstract contrasting the different conceptual models. The Introduction suffers from the same problem. While most readers familiar with mountain hydrological processes will know the typical range for MBR and MFR, it may be useful for the larger HESS community to provide numbers (range of recharge) with geographic setting. This is easy to incorporate into a table and provides the readers with what to expect for MBR and MFR, why it's important, etc. The Conclusions is the weakest section in my opinion and may need a complete overhaul. The article is entertaining and will likely gain appreciation in the communities mentioned above. However, the Conclusions are methods-heavy and do not discuss the regional or broader implications.

Second, Figures 6 and 7 are difficult to interpret. Figure 8 is much more informative. Is there a way to show this data more effectively? This is important since so much of the interpretation revolves around these figures.

Third, it may be very useful/informative to have conceptual diagrams for Section 2.1, especially for those readers who have not read Winter's seminal chapter.

Specific Comments: P2 L2 (Page 2 Line 2): suggest changing "rainfall" to either "precipitation" or "rainfall and snowfall" for the broader audience. Suggest adding references to Winograd et al. (1998) and Earman et al. (2006).

P2 L23: suggest changing "ultimate" to "most robust" or "more robust".

P3 L3: it's still uncommon to have many wells in other mountainous regions. Perhaps

this statement should be tempered to reflect regional conditions.

P3 L6-9: Can you expand on what is implied in this statement? It assumes that the reader has an intimate knowledge of the references studies.

P3 L23: What is meant by "triangular facets"? Do you mean interfluve? Please define.

P4 L6: This is not correct. It should be groundwater discharge to streams occurs when groundwater levels are higher than the stream as indicated in Winter's chapter. Correct?

P4 L22: delete "as discussed in the following" and elsewhere in the article.

P5 L18-20: This is not always the case, not always true.

P5 L21: This is a sweeping statement. Is it possible that Cl-(MFR) can be higher than Cl-(MBR), especially where streams draining the mountains are accumulating lots of Cl- from groundwater in other regions?

P6 L11-18: Is there seasonality in precipitation and recharge?

P10 L9-14: please add arrows to show these locations.

P11 L2: suggest amending the statement to read, "had the same Cl concentrations or if mineral sources of Cl were non-negligible and/or spatially variable, "

P11 L11: should this be "long" instead of "short"?

P13 L27-30: do you have Br data? If so, Cl/Br ratios would provide additional support for this statement.

---

## Referee Comment (RC2) · Anonymous Referee #2 · 5 Oct 2017

This manuscript uses chloride concentrations (and electrical conductivity data as a proxy for chloride concentrations) along with hydraulic head data to examine the relative importance of mountain-front recharge vs. mountain-block recharge in aquifers near the Mount Lofty Ranges in Australia. Overall I think that this manuscript is good, and the method they are using is something that can be applied in other basins in a relatively simple, cost-effective manner. However, there are some areas where I feel they need to strengthen some of the foundations—in some cases I am not able to see in the figures what they describe, and there are some alternative views I have expressed. I hope the authors are able to address these comments, as I think it would result in a very good paper.

**'Scientific' comments:**

p 2 L 3: you mention mountains receive higher rainfall, but another important consideration is that in many parts of the world, mountains are high enough in elevation that they receive snow when surrounding low-lying areas receive rain. Several studies (e.g., Earman et al., 2006; Simpson et al., 1972; and Winograd et al., 1998) have shown that snow can be a much more 'efficient' recharge agent than rain (e.g., the percentage of recharge derived from snowmelt exceeds the portion of precipitation that falls as snow). A quick web search appears to show that your study area receives snow at least in some years; even if it did not, if you wish your case study to have broad applicability, you should mention this issue.

p 2 L 28 (and numerous other instances): since you are referring to the chloride ion (not elemental chlorine [see comment p 5 L 5], you should use the symbol "$Cl^-$", not "$Cl$"

p 3 L 23-24: can you show a diagram of what these 'triangular facets' are? It is not clear to me from the text.

p 4 L 25: I'm not sure what is referred to by 'embedding materials'—do you mean the fault has the same K as the basement; the same K as the sediment; or that the upper half of the fault has the same K as the sediment, and the lower half of the fault has the same K as the basement?

p 5 L 5: this sentence needs to be rewritten to be chemically correct. *Chloride* is **not** an element—it is the monovalent anionic form of the element *chlorine*. Chlorine [the element] is quite highly reactive, not "relatively non-reactive compared to the other elements". The chloride ion is usually considered to be a 'conservative' tracer in groundwater systems, but you are confusing the issue by referring to chloride as an element.

p 5 L 8: here it is stated that $Cl^-$ should remain equal to the concentration at the time of recharge if dispersion can be neglected. Dispersion is likely to be the only process that can reasonably be expected to *reduce* $Cl^-$ concentration under most hydrogeologic conditions (precipitation of $Cl^-$-bearing minerals will only take place at extremely high concentrations, typically after a significant [>50% of original water] evaporation). However, there are many factors that could *increase* $Cl^-$ concentration (e.g., salting of roads in the area, mixing with brine, dissolution of

evaporite minerals, impacts from fertilizer, influence of septic/sewage systems, irrigation). You may be able to rule those out in your study area, but those influences should at least be mentioned.

p 5 L 11: atmospheric deposition is often the main influence on groundwater chloride, but there are other natural and anthropogenic sources of chloride (see previous comment)

p 5 L 14: I understand that evaporation leaves residual water more concentrated in all ionic species, but the impact of 'transpiration' on groundwater ionic composition has me a bit puzzled. I have found many references that say 'transpiration' removes pure water (leaving the residual enriched in ions), but those references all seem to refer to 'transpiration' as evaporation from the stomata in plant leaves. So I am left wondering if the actual uptake of water by plants at the root actually removes pure water or water and ions. My guess is that the uptake of soil water or groundwater into a plant root is controlled simply by head gradient, in which case ions should move into the plant along with water. If that is correct, water in plant leaves may become enriched in ions, but the uptake of the water itself would not be enriching [I admit that this is completely outside my area of expertise, so this assumption could be dead wrong!].

p 5 L 27-29: you refer to the need for reducing potential misinterpretation due to spatial variability, but on lines 15-16, you said that chloride concentrations in mountains "can be expected to show significant spatio-temporal variability", so shouldn't you be concerned with temporal variability in addition to spatial variability?

p 6 L 19-p 7 L 11: is it possible to include at least a conceptual schematic diagram showing a typical cross section of these aquifers?

p 7 L 6: you mention that wells in the area are used for "irrigation and industrial activities". Related to comment (p 5 L 8), I will point out that irrigation is often a driver of increased groundwater salinity, and industrial uses can also increase salinity; perhaps you should discuss potential impacts of these activities on the CMB method.

p 7 L 7: suggest replacing "permanent" with "long-term". Perhaps I'm being overly pedantic, but these cones of depression are 'permanent' only as long as water use remains higher than recharge. I realize that you say water use is forecast to rise instead of drop, but it is still possible (although unlikely given societal constraints) to make the cones of depression go away.

p 8 L 32-p 9 L 1: I don't think I agree with this statement. If you were discussing the rate of water chemistry change due to chemical reactions (e.g., weathering, dissolution, etc.), I would agree that chemical concentrations may often change relatively slowly compared to groundwater flow (even though groundwater flow can often be slow). However, pumping can induce chemical mixing by changing hydraulic gradient and bringing 'new' water into the pumping zone. For instance, some of the $Cl^-$ values you report (Figure 5) are extremely high (since the dots overlap in Figure 5, it's hard to estimate how many samples are there, but a decent chunk of

your samples have $Cl^-$ > 1,000 mg/L (topping out with a $Cl^-$ value around $10^5$ mg/L!), which is quite high (I'm guessing these might be paleowaters in the lower aquifer). If, for example, pumping caused increased interaquifer flow, the chemical change brought about by mixing would occur on the same time scale as the hydraulic mixing.

p 9 L 1: here, you state "Hence, as for hydraulic heads, all available Cl data were retained"; I'm having trouble resolving that with your explanation of your hydraulic head data set in section 3.2.1, especially p 7 L 26 where you state "the data were filtered out for unsuitable measurements…" and p 7 L 27 where you state "After filtering, 111,538 hydraulic head measurements from 9,561 wells were obtained."

Section 3.3.1: in this section, you discuss the relationship between head contours and streams, and use it to invoke flow into/out of the streams (e.g., streams are gaining in location *x*, but losing in zone *y*). One the scale of Figure 7, I can't really see that. Is it possible to show one or two 'details' of head contours near a stream, similar to Figure 8B and 9B in Winter et al. (1998)? If you could show something similar to those figures, I think it would greatly strengthen your argument.

p 10 L 5-8: You state that with the exception of the upper reaches of rivers, head contours indicate streams in the Mount Lofty Ranges are gaining (lines 5-6), yet on line 8, you state that "the infiltration capacity of the mountain block is limited". If the mountain block has such limited infiltration capacity, where is the water that causes the streams to be gaining coming from? If the streams are gaining, I assume it's because groundwater is flowing in, which would suggest the mountain block must be capable of receiving/transmitting water fairly well.
The root of your 'low infiltration capacity' hypothesis seems to be on lines 6-7, where you observe that the principal exception to the 'streams are typically gaining in the Mount Lofty Ranges' rule is "along the upper reaches of rivers, i.e. where the stream order is small". A few thoughts on the upper reaches of streams:

> 1. Are you sure the upper-reach sections of stream that you are discussing here are perennial? I have spent a lot of time in mountains, and a lot of that time has been spent sampling from springs. In my experience, virtually all perennial mountain streams in non-glaciated regions originate as groundwater outflow. If, for example, you are relying on a GIS data set for stream locations/origination points, your data set may have sections of stream channel that are ephemeral digitized in (e.g., from air photos) with no distinction from perennial reaches of stream. By definition, an ephemeral stream channel will be reliant on overland flow/interflow after precipitation instead of groundwater. Have you ground-truthed any of these upper reaches of stream to make sure they are perennial? If the upper reaches that don't appear to be gaining are ephemeral, it removes support for your 'limited infiltration capacity' hypothesis (although as already stated, the fact that the rest of the stream reaches are gaining already goes against that hypothesis).

2. Upper reaches of rivers in mountains tend to be in the highest-elevation areas, where wells (and thus head values) are typically the sparsest. Do you have the data density in the zones surrounding the upper reaches of rivers in the Mount Lofty Ranges to make the shapes/locations of your head contours definitive enough to truly tell whether the streams are gaining or losing? Looking at the head data points relative to stream headwaters in Figure 7, I'm not sure you have the data density to make this call. Also, looking at Figure 7, I'm not really able to see head contours 'veeing' (as Winter et al. (1998) show to indicate gaining or losing streams [depending on 'vee' direction relative to upstream/downstream]) to/from streams in most instances. If you have the data density to make plots as suggested in my comment on Section 3.3.1 and actually show the 'veeing', that would be much stronger support of your case.

3. As mentioned in (2), upper reaches of rivers in mountains tend to be in the highest-elevation areas, which means they would tend to be recharge zones, not discharge zones. Your Figure 1 suggests that mountaintops are recharge zones, and that it is necessary to get some distance downslope before you reach a groundwater discharge zone. This is a potential argument against the idea that "the infiltration capacity of the mountain block is limited". Recharge zones can have very high infiltration capacity, but because they are recharge zones, we don't expect discharge to occur. As a result, I don't believe that lack of discharge alone can be used as a test for low infiltration capacity. The upper stream reaches may be getting no groundwater inflow because they are in recharge zones instead of discharge zones.

To sum up, I'm not sure the infiltration capacity of the mountain block is limited, but if you can present evidence more likely to convince me, I'd be willing to reconsider!

p 12 L 7-8: see my earlier comment (p 10 L 5-8) that I think it is likely perennial streamflow in your system is generated by groundwater inflow, not by overland flow/interflow (also see your statement on line 19 of this page)

p 13 L 11-13: see comment p 8 L 32-p 9 L 1; if pumping induces mixing between waters with different chloride concentrations, the chemical change should occur at the same rate as the hydraulic change

p 14 L 11: this could probably use some clarification—I think you are describing using an airlift pump at the same pressure in multiple wells and assigning higher T values to wells that yield a relatively high amount of water and lower T values to wells that yield a relatively low amount of water. Please note that many readers might be unfamiliar with airlift pumps, so this could merit a bit more explanation than you currently give.

p 11 L 19: what are you defining as "the front line"? Is it the fault that mostly runs along the eastern boundary of the NAP (but continues on through the CAP), is it the fault that mostly runs along the eastern boundary of the CAP, is it somewhere else? This should be clarified.

p 14 L 30-33: at the end of this sentence, you refer only to 'water found in streams running down the mountain' to contrast to water that recharges diffusively, but I think the important concept here is that [some/much of] the water flowing in the streams will become focused recharge in contrast to the diffuse recharge found outside stream zones. I think you need to make clear to the reader that the important concept here is diffuse vs. focused recharge.

Conclusions: I will have an easier time agreeing with all the conclusions if you can address some of the earlier comments I made (e.g., if you can make figures as suggested in my comment on Section 3.3.1, I can better believe that analysis of head contour shape adjacent to surface features allows you to distinguish gaining/losing reaches of streams).

**Grammar/spelling/etc. comments & comments on figures/tables:**

Title: should read "electrical-conductivity data"

p 2 L 16 (and numerous other instances): not sure of journal style, but typically "e.g." is followed by a comma, e.g., "e.g., Hely et al., 1971…"

p 3 L 4: you make reference to a geographic location here, so you should 'call' a figure that shows that location. Your Figure 3 shows this area, but you need to move the call to this first mention of the area in the text (which will cause it to become Figure 2)

p 3 L 23 (and any other instances): not sure of journal style, but typically "i.e." is followed by a comma, e.g., "i.e., the in-between-streams zones…"

p 4 L 1: Winter et al., 1998 is cited here (and on line 8), but does not appear in the reference list, need to add a reference.

p 5 L 32: change "increasing" to "increase"

p 7 L 27: change "aquifer test or drilling" to either "aquifer tests or drilling" or "aquifer testing or drilling"

p 8 L 4: sentence is unclear—I'm not sure what is meant by the statement that the aquifer "was informed in the database for about two thirds of the wells".

p 8 L 27: change "less restrictive filtering" to "less-restrictive filtering"

Figure 6: two issues with this figure:

1. many of the contour labels are very difficult to read (too small); especially the red/pink labels for topography (and those are even worse over some of the areas with dark red-brown fill in the southeastern edge of the map).
2. you do not show a key for the contour fill colors that you use, if you provided one, that might help with problem (1)

p 10 L 15: change "by" to "minus"

Figure 10: the faults are often difficult to see (especially since one is nearly the same color as the CAP boundary (but a thinner line than the CAP boundary), and runs on/very near the CAP boundary for a good distance. Could the faults be some other color on this figure (the green of the Australian 'green and gold' color scheme might be one choice that would stand out a bit better)?

p 11 L 20: change "occur" to "occurs"

p 11 L 25: cut the first instance of "salinity" on this line

p 11 L 28: change "do not contribute either" to "also do not contribute"

Figure 11/Table 1:
   1. In Figure 11, you plot flow rate using the units ML/d, but in Table 1, you report mean flow rates in GL/y, which makes comparing the two difficult. Please pick one consistent set of units for flow rate/mean flow rate and use it in both Figure 11 and Table 1.

   2. At first glance, there appear to be two sets of 'paired' streams (e.g., the symbols used for South Para River and First Creek are identical to my eye, as are the symbols for Brownhill Creek and Gawler River). On closer examination, I can see a slight difference between the Grawler/Brownhill point colors, but it is tough to tell which is which (I think Brownhill is the lower-concentration data set on the plot); I can't make any difference out for the other 'pair'). Can you do something to make each data set more distinct? One suggestion: there are only so many colors you can use that work well unoutlined, but if your graphing software is able to outline points, they can become much more 'readable' (e.g., yellow circles on their own can be hard to see on a white background, but a yellow circle with a black border shows up well on a white background). Using outlined shapes might let you add a couple of distinct colors that would make it much easier to tell the data sets apart on the figure.

   3. In Table 1, change two instances of "uS/cm" to "μS/cm"

p 13 L 1: change "great" to something along the lines of "useful", "effective", etc.

p 14 L 15: cut "based on"

p 14 L 23: cut "only"

p 14 L 25: add comma after 'contrast'

p 14 L 28: cut "yet"

p 15 L 1-2: change "this study proposes" to "we propose" [or if the journal style does not allow that, the more stilted "the authors propose"].  Your study can't propose anything, but you can!

**References:**

Earman, S., Campbell, A. R., Newman, B. D., and Phillips, F. M., 2006. Isotopic exchange between snow and atmospheric water vapor: Estimation of the snowmelt component of groundwater recharge in the southwestern United States, *Journal of Geophysical Research*, 111: D09302, doi:10.1029/2005JD006470.

Simpson, E. S., Thorud, D. B., and Friedman, I., 1972. Distinguishing seasonal recharge to groundwater by deuterium analysis in southern Arizona. In: *World Water Balance. Proceeding of the Reading Symposium, July 1970*, Volume 3, International Association of Scientific Hydrology-UNESCO-WMO Studies and Reports in Hydrology, vol.11; Gentbrugge, Belgium, Publication No 94 of the International Association of Scientific Hydrology, pp 623–633

Winograd, I. J., Riggs, A. C.. and Coplen, T. B., 1998. The relative contributions of summer and cool-season precipitation to groundwater recharge, Spring Mountains, Nevada, USA, *Hydrogeology Journal* (6) 77– 93.

Winter T. C., Harvey, J. W., Franke, O. L., and Alley, W. M., 1998. *Groundwater and Surface Water: A Single Resource*, U.S. Geological Survey Circular 1139.

---

## Author Comment (AC1) · 10 Nov 2017

**Response to Anonymous Referee #1:**

Comments in black

Responses in blue

General Comments:

This manuscript presents research that was completed to determine the source(s) of recharge to the Adelaide Plains, Australia. The authors were specifically interested in determining whether recharge in the Adelaide Plains was sourced from mountain-block recharge (MBR) or mountain-front recharge (MFR) or a combination of the two in the nearby Mount Lofty Range using a combination of regional hydraulic head data and chloride data. The article is well written and will likely gain broad readership within the hydrogeology and mountain hydrology communities. It has immediate regional implications to the evolving conceptual framework for the Adelaide Plains.

There are a couple of weaknesses in its current form that prevent a recommendation for acceptance.

First, the Abstract, Introduction, and Conclusions do not do the article justice. The Abstract needs data in my opinion, but this is admittedly difficult since most of the results are presented spatially in maps. In any case, the Abstract feels methods-heavy since only the last 6 sentences discuss actual results. This could be remedied with a concise problem statement in the Abstract contrasting the different conceptual models. The Introduction suffers from the same problem. While most readers familiar with mountain hydrological processes will know the typical range for MBR and MFR, it may be useful for the larger HESS community to provide numbers (range of recharge) with geographic setting. This is easy to incorporate into a table and provides the readers with what to expect for MBR and MFR, why it's important, etc. The Conclusions is the weakest section in my opinion and may need a complete overhaul. The article is entertaining and will likely gain appreciation in the communities mentioned above. However, the Conclusions are methods-heavy and do not discuss the regional or broader implications.

Response: We agree that these sections may contain too many technical details, and so we will remove some of these these while revising the manuscript, and will put more emphasis on the implications. However, we prefer to keep the focus principally on the methods, as this is what the article is about. Namely, including the typical range for MBR and MFR is deemed out of scope as this would not serve the paper's objective, and considering that the recharge rate would be hugely dependent on regional characteristics (climate, geology, topography).

Second, Figures 6 and 7 are difficult to interpret. Figure 8 is much more informative. Is there a way to show this data more effectively? This is important since so much of the interpretation revolves around these figures.

Response: We understand the concern. To deal with this issue, we will consider producing zoomed-in figures when revising the manuscript.

Third, it may be very useful/informative to have conceptual diagrams for Section 2.1, especially for those readers who have not read Winter's seminal chapter.

Response: Accepted; we will add a figure.

Specific Comments:

P2 L2 (Page 2 Line 2): suggest changing "rainfall" to either "precipitation" or "rainfall and snowfall" for the broader audience. Suggest adding references to Winograd et al. (1998) and Earman et al. (2006).

Response: Accepted (will use "precipitation").

P2 L23: suggest changing "ultimate" to "most robust" or "more robust".

Response: Accepted ("most robust").

P3 L3: it's still uncommon to have many wells in other mountainous regions. Perhaps this statement should be tempered to reflect regional conditions.

Response: Accepted.

P3 L6-9: Can you expand on what is implied in this statement? It assumes that the reader has an intimate knowledge of the references studies.

Response: We will clarify the statement by changing it for: "In particular, the hydraulic role of faults (i.e. acting as barrier or conduit to flow) that run along the mountain front remains unclear."

P3 L23: What is meant by "triangular facets"? Do you mean interfluve? Please define.

Response: We will add a figure to clarify this concept.

P4 L6: This is not correct. It should be groundwater discharge to streams occurs when groundwater levels are higher than the stream as indicated in Winter's chapter. Correct?

Response: Thank you for pointing out the error; this will be corrected.

P4 L22: delete "as discussed in the following" and elsewhere in the article.

Response: Accepted.

P5 L18-20: This is not always the case, not always true.

Response: Accepted; we will mention this as only one possibility.

P5 L21: This is a sweeping statement. Is it possible that Cl-(MFR) can be higher than Cl-(MBR), especially where streams draining the mountains are accumulating lots of Cl- from groundwater in other regions?

Response: Accepted; we will indicate this possibility.

P6 L11-18: Is there seasonality in precipitation and recharge?

Response: Yes. We will add this information, as in: "The majority (87 %) of the rainfall in the Mount Lofty Ranges occurs during the extended winter season (April–September) (station number 23810, 1970–2013), indicating a strong seasonality in the recharge.".

P10 L9-14: please add arrows to show these locations.

Response: We will consider producing zoomed-in figures and adding interpretation information when revising the manuscript.

P11 L2: suggest amending the statement to read, "had the same Cl concentrations or if mineral sources of Cl were non-negligible and/or spatially variable, "

Response (P13 and not P11): We prefer to modify the sentence as: "Cl is potentially a great tool to distinguish between MFR and MBR if the stream water Cl concentration in the mountain front zone is significantly different from the groundwater Cl concentration at the base of the mountain.". This sentence is more rigorously correct, while the question of mineral sources is addressed later.

P11 L11: should this be "long" instead of "short"?

Response (P13 and not P11): Absolutely; thank you for pointing out the error. This will be corrected.

P13 L27-30: do you have Br data? If so, Cl/Br ratios would provide additional support for this statement.

Response: We will look into that while revising the manuscript.

---

## Author Response (AR1)

**Response to Anonymous Referee #1:**

Comments in black

Responses in blue

General Comments:

This manuscript presents research that was completed to determine the source(s) of recharge to the Adelaide Plains, Australia. The authors were specifically interested in determining whether recharge in the Adelaide Plains was sourced from mountain-block recharge (MBR) or mountain-front recharge (MFR) or a combination of the two in the nearby Mount Lofty Range using a combination of regional hydraulic head data and chloride data. The article is well written and will likely gain broad readership within the hydrogeology and mountain hydrology communities. It has immediate regional implications to the evolving conceptual framework for the Adelaide Plains.

There are a couple of weaknesses in its current form that prevent a recommendation for acceptance.

First, the Abstract, Introduction, and Conclusions do not do the article justice. The Abstract needs data in my opinion, but this is admittedly difficult since most of the results are presented spatially in maps. In any case, the Abstract feels methods-heavy since only the last 6 sentences discuss actual results. This could be remedied with a concise problem statement in the Abstract contrasting the different conceptual models. The Introduction suffers from the same problem. While most readers familiar with mountain hydrological processes will know the typical range for MBR and MFR, it may be useful for the larger HESS community to provide numbers (range of recharge) with geographic setting. This is easy to incorporate into a table and provides the readers with what to expect for MBR and MFR, why it's important, etc. The Conclusions is the weakest section in my opinion and may need a complete overhaul. The article is entertaining and will likely gain appreciation in the communities mentioned above. However, the Conclusions are methods-heavy and do not discuss the regional or broader implications.

Response: We agree that these sections may contain too many technical details. We removed some of these and put more emphasis on the implications. However, we preferred to keep the focus principally on the methods, as this is what the core of the article is about. Namely, including the typical range for MBR and MFR is deemed out of scope as this would not serve the paper's objective. The conclusion was completely rewritten.

Second, Figures 6 and 7 are difficult to interpret. Figure 8 is much more informative. Is there a way to show this data more effectively? This is important since so much of the interpretation revolves around these figures.

Response: We understand the concern. To deal with this issue, we made zoomed-in figures (Figures 9-11 in the revised manuscript). Figures covering the whole area are now available in supplementary material (Figures S1-S12).

Third, it may be very useful/informative to have conceptual diagrams for Section 2.1, especially for those readers who have not read Winter's seminal chapter.

Response: Accepted; we added a figure (Figure 4).

Specific Comments:

P2 L2 (Page 2 Line 2): suggest changing "rainfall" to either "precipitation" or "rainfall and snowfall" for the broader audience. Suggest adding references to Winograd et al. (1998) and Earman et al. (2006).

Response: Accepted (we used "precipitation").

P2 L23: suggest changing "ultimate" to "most robust" or "more robust".

Response: Accepted (we used "most robust").

P3 L3: it's still uncommon to have many wells in other mountainous regions. Perhaps this statement should be tempered to reflect regional conditions.

Response: Accepted. This general sentence was deleted and the fact that it is the case for the study area has been indicated later (P3 L10-11).

P3 L6-9: Can you expand on what is implied in this statement? It assumes that the reader has an intimate knowledge of the references studies.

Response: We clarified the statement by changing it for: "In particular, the hydraulic role of faults (i.e. acting as barrier or conduit to flow) that run along the mountain front remains unclear."

P3 L23: What is meant by "triangular facets"? Do you mean interfluve? Please define.

Response: We expanded the text and added a figure to clarify this concept.

P4 L6: This is not correct. It should be groundwater discharge to streams occurs when groundwater levels are higher than the stream as indicated in Winter's chapter. Correct?

Response: Thank you for pointing out the error; this has been corrected.

P4 L22: delete "as discussed in the following" and elsewhere in the article.

Response: Accepted.

P5 L18-20: This is not always the case, not always true.

Response: Accepted; we changed the text for mentioning different possibilities.

P5 L21: This is a sweeping statement. Is it possible that Cl-(MFR) can be higher than Cl-(MBR), especially where streams draining the mountains are accumulating lots of Cl- from groundwater in other regions?

Response: Accepted; we emphasized in the text that "no general statement can be made".

P6 L11-18: Is there seasonality in precipitation and recharge?

Response: Yes; this information was added (P6 L28-30): "The majority (77 %) of the rainfall in the Mount Lofty Ranges occurs during autumn and winter (May–September) (station number 23810, 1970–2013), suggesting a strong seasonality in the recharge.".

P10 L9-14: please add arrows to show these locations.

Response: We made zoomed-in figures and added arrows indicating groundwater flow directions to make it easier to read these figures.

P11 L2: suggest amending the statement to read, "had the same Cl concentrations or if mineral sources of Cl were non-negligible and/or spatially variable, "

Response (comment interpreted as on P13 and not P11): This was in fact incorporated by the fact that by sources we meant stream water at the mountain front (for MFR) and groundwater at the base of the mountain (for MBR). The underlying mechanisms for different Cl concentrations in these sources are discussed in section 2.2. Nonetheless, to clarify, we revised as (P13, L25-28): "But for this, the stream water Cl⁻ concentration in the mountain front zone (i.e., potential MFR source) needs to be significantly different from the groundwater Cl⁻ concentration at the base of the mountain (i.e., potential MBR source). Different processes involved in the generation of MFR and MBR imply that these two potential sources of water may indeed have different Cl⁻ concentrations (see section 2.2)."

P11 L11: should this be "long" instead of "short"?

Response (comment interpreted as on P13 and not P11): Absolutely; thank you for pointing out the error. The sentence has been changed for "However, changes in solute concentrations are expected to be observed later than hydraulic head changes".

P13 L27-30: do you have Br data? If so, Cl/Br ratios would provide additional support for this statement.

Response: We analysed Br data and incorporated the results in the manuscript (P14 L20-25 and Figure S13): "This assumption was also tested by analysing the chloride/bromide (Cl⁻/Br⁻) ratio from 161 well samples distributed over the study area. The average Cl⁻/Br⁻ ratio from these samples is 739 +/- 173 (molar) (Figure S13). This shows that groundwater has a similar Cl⁻/Br⁻ ratio to seawater (~650 molar), hence indicating that there is little water-rock interactions or dissolution of evaporite deposits (Drever, 1997; Davis et al., 1998). The low variance of the Cl⁻/Br⁻ ratio also suggests that the dominant process for the increasing salinity in the groundwater is evapotranspiration (Cartwright et al., 2006).".

**Response to Anonymous Referee #2:**

Comments in black

Responses in blue

This manuscript uses chloride concentrations (and electrical conductivity data as a proxy for chloride concentrations) along with hydraulic head data to examine the relative importance of mountain-front recharge vs. mountain-block recharge in aquifers near the Mount Lofty Ranges in Australia. Overall I think that this manuscript is good, and the method they are using is something that can be applied in other basins in a relatively simple, cost-effective manner. However, there are some areas where I feel they need to strengthen some of the foundations—in some cases I am not able to see in the figures what they describe, and there are some alternative views I have expressed. I hope the authors are able to address these comments, as I think it would result in a very good paper.

'Scientific' comments:

p 2 L 3: you mention mountains receive higher rainfall, but another important consideration is that in many parts of the world, mountains are high enough in elevation that they receive snow when surrounding low-lying areas receive rain. Several studies (e.g., Earman et al., 2006; Simpson et al., 1972; and Winograd et al., 1998) have shown that snow can be a much more 'efficient' recharge agent than rain (e.g., the percentage of recharge derived from snowmelt exceeds the portion of precipitation that falls as snow). A quick web search appears to show that your study area receives snow at least in some years; even if it did not, if you wish your case study to have broad applicability, you should mention this issue.

Response: Accepted; the term "rainfall" has been changed for "precipitation" such as to encompass snowfall. We also added references to Earman et al. (2006) and Winograd et al. (1998). Later in the text, we also mentioned the insignificance of snow in the case study area (P6 L21 and L24-25).

p 2 L 28 (and numerous other instances): since you are referring to the chloride ion (not elemental chlorine [see comment p 5 L 5], you should use the symbol "Cl-", not "Cl"

Response: Accepted.

p 3 L 23-24: can you show a diagram of what these 'triangular facets' are? It is not clear to me from the text.

Response: Accepted; we added a figure to clarify this concept (Figure 3).

p 4 L 25: I'm not sure what is referred to by 'embedding materials'—do you mean the fault has the same K as the basement; the same K as the sediment; or that the upper half of the fault has the same K as the sediment, and the lower half of the fault has the same K as the basement?

Response: It is simply assumed that the fault only implies a difference in basement elevation and no alteration in hydraulic conductivity; this has been clarified in the text (P4 L26-27): "The hydraulic conductivity of the fault zone itself is not different from that of the embedding materials (i.e., the fault only implies a difference in basement elevation).".

p 5 L 5: this sentence needs to be rewritten to be chemically correct. Chloride is not an element—it is the monovalent anionic form of the element chlorine. Chlorine [the element] is quite highly reactive, not "relatively non-reactive compared to the other elements". The

chloride ion is usually considered to be a 'conservative' tracer in groundwater systems, but you are confusing the issue by referring to chloride as an element.

Response: Accepted; this has been corrected.

p 5 L 8: here it is stated that Cl- should remain equal to the concentration at the time of recharge if dispersion can be neglected. Dispersion is likely to be the only process that can reasonably be expected to reduce Cl- concentration under most hydrogeologic conditions (precipitation of Cl--bearing minerals will only take place at extremely high concentrations, typically after a significant [>50% of original water] evaporation). However, there are many factors that could increase Cl- concentration (e.g., salting of roads in the area, mixing with brine, dissolution of evaporite minerals, impacts from fertilizer, influence of septic/sewage systems, irrigation). You may be able to rule those out in your study area, but those influences should at least be mentioned.

Response: Accepted; we took on this comment by adding the following sentence: "Other potential sources of Cl$^-$ in groundwater include anthropogenic inputs (e.g., salting of roads, irrigation, application of fertilizer, leakage from septic/sewage systems) and dissolution of Cl-bearing minerals. Cl$^-$ removal from solution is unlikely as Cl does not easily adsorb onto clays or precipitate as mineral (Clark and Fritz, 1997).".

p 5 L 11: atmospheric deposition is often the main influence on groundwater chloride, but there are other natural and anthropogenic sources of chloride (see previous comment)

Response: Accepted (see our answer to the previous comment).

p 5 L 14: I understand that evaporation leaves residual water more concentrated in all ionic species, but the impact of 'transpiration' on groundwater ionic composition has me a bit puzzled. I have found many references that say 'transpiration' removes pure water (leaving the residual enriched in ions), but those references all seem to refer to 'transpiration' as evaporation from the stomata in plant leaves. So I am left wondering if the actual uptake of water by plants at the root actually removes pure water or water and ions. My guess is that the uptake of soil water or groundwater into a plant root is controlled simply by head gradient, in which case ions should move into the plant along with water. If that is correct, water in plant leaves may become enriched in ions, but the uptake of the water itself would not be enriching [I admit that this is completely outside my area of expertise, so this assumption could be dead wrong!].

Response: Indeed, plant roots do not uptake pure water, as plants need nutrients, including Cl (White and Broadley, 2001). Subsequently, Cl is either consumed by the plant (i.e. it enters its composition) or it is released again by leaching. The exact mechanisms of Cl uptake are complex, but it does not matter for our purpose as long as Cl does not go back to the atmosphere. An assumption is needed though: that vegetation is in equilibrium, so that it has no net effect on the chloride flux reaching groundwater (i.e., only the water flux is reduced due to transpiration, and so the Cl concentration is indeed increased). To be sure, the assumption of equilibrium of the vegetation was added, and another supporting reference for the increase of chloride concentration as a result of evapotranspiration was also added (Allison et al., 1994). The text now reads (P5 L17-21): "The Cl$^-$ concentration in recharge water also depends on evapotranspiration, which leaves Cl$^-$ in solution, implying its enrichment (Eriksson and Khunakasem, 1969; Allison et al., 1994) (a growing vegetation could in theory counter this effect since Cl is a nutrient for plants (White and Broadley, 2001), but in practice the uptake of Cl from soil by most plant species is insignificant (Allison et al., 1994)).".

*References:*

White, P. J., and Broadley, M. R.: Chloride in Soils and its Uptake and Movement within the Plant: A Review, Annals of Botany, 88, 967-988, 2001.

Allison, G. B., Gee, G. W., and Tyler, S. W.: Vadose-zone techniques for estimating groundwater recharge in arid and semiarid regions, Soil Science Society of America Journal, 58, 6-14, 1994.

p 5 L 27-29: you refer to the need for reducing potential misinterpretation due to spatial variability, but on lines 15-16, you said that chloride concentrations in mountains "can be expected to show significant spatio-temporal variability", so shouldn't you be concerned with temporal variability in addition to spatial variability?

Response: True; in fact, as much as possible, the analysis should focus on comparing points that are not far apart from another and along potential flow paths, such as to reduce risks of misinterpretation caused by potentially both spatial and temporal variability in Cl$^-$ concentration and dispersion. This was clarified in the following sentences (P6 L1-5): "As much as possible, the analysis should focus on comparing points that are not too far apart from another and along presumed flow paths, such as to reduce risks of misinterpretation caused by spatiotemporal variability in Cl$^-$ concentration and dispersion. In particular, MBR should be most reliably assessed by comparing the groundwater Cl$^-$ concentration in the uppermost part of the basin to that in the lowermost part of the mountain, along lines running perpendicular to the mountain front."

p 6 L 19-p 7 L 11: is it possible to include at least a conceptual schematic diagram showing a typical cross section of these aquifers?

Response: Accepted; we added a figure (Figure 7).

p 7 L 6: you mention that wells in the area are used for "irrigation and industrial activities". Related to comment (p 5 L 8), I will point out that irrigation is often a driver of increased groundwater salinity, and industrial uses can also increase salinity; perhaps you should discuss potential impacts of these activities on the CMB method.

Response: Accepted; we mentioned this potential effect in two places. P9 L16: "Additionally, in the uppermost aquifers, irrigation may have locally influenced the Cl$^-$ concentration.". P14 L1-4: "In addition, irrigation may have influenced the Cl$^-$ concentration in shallow wells located in agricultural areas. However, changes in solute concentrations are expected to be observed later than hydraulic head changes, and a dramatic shift in Cl$^-$ concentrations is unlikely to be seen in Cl$^-$ concentrations away from the main areas of pumping.".

p 7 L 7: suggest replacing "permanent" with "long-term". Perhaps I'm being overly pedantic, but these cones of depression are 'permanent' only as long as water use remains higher than recharge. I realize that you say water use is forecast to rise instead of drop, but it is still possible (although unlikely given societal constraints) to make the cones of depression go away.

Response: Accepted.

p 8 L 32-p 9 L 1: I don't think I agree with this statement. If you were discussing the rate of water chemistry change due to chemical reactions (e.g., weathering, dissolution, etc.), I would agree that chemical concentrations may often change relatively slowly compared to groundwater flow (even though groundwater flow can often be slow). However, pumping can induce chemical mixing by changing hydraulic gradient and bringing 'new' water into the

pumping zone. For instance, some of the Cl- values you report (Figure 5) are extremely high (since the dots overlap in Figure 5, it's hard to estimate how many samples are there, but a decent chunk of your samples have Cl- > 1,000 mg/L (topping out with a Cl- value around 105 mg/L!), which is quite high (I'm guessing these might be paleowaters in the lower aquifer). If, for example, pumping caused increased interaquifer flow, the chemical change brought about by mixing would occur on the same time scale as the hydraulic mixing.

Response: We reiterate that the influence on hydraulic head would typically be seen faster than the influence on chemistry, since the former refers to pressure travel time while the latter refers to solute travel time. We made it clearer in the revised manuscript as in: "this effect may be seen later than on hydraulic heads since solutes travel times are typically longer than pressure travel times". Regarding the high Cl⁻ concentrations, these are not necessarily due to pumping: many (in fact, most) of these come from observation wells that are not necessarily close to pumping wells but from parts of the aquifer system which have a paleo-water composition or are subject to high evapotranspiration.

p 9 L 1: here, you state "Hence, as for hydraulic heads, all available Cl data were retained"; I'm having trouble resolving that with your explanation of your hydraulic head data set in section 3.2.1, especially p 7 L 26 where you state "the data were filtered out for unsuitable measurements…" and p 7 L 27 where you state "After filtering, 111,538 hydraulic head measurements from 9,561 wells were obtained."

Response: This has been clarified by changing this sentence into: "Nonetheless, following the same reasoning as for hydraulic heads, all available Cl- data were retained regardless of the measurement date".

Section 3.3.1: in this section, you discuss the relationship between head contours and streams, and use it to invoke flow into/out of the streams (e.g., streams are gaining in location x, but losing in zone y). One the scale of Figure 7, I can't really see that. Is it possible to show one or two 'details' of head contours near a stream, similar to Figure 8B and 9B in Winter et al. (1998)? If you could show something similar to those figures, I think it would greatly strengthen your argument.

Response: We understand the concern and so we made zoomed-in figures (Figures 9-11 in the revised manuscript). We also added arrows indicating groundwater flow directions to make it easier to read these figures. Figures covering the whole area are now available in supplementary material (Figures S1-S12).

p 10 L 5-8: You state that with the exception of the upper reaches of rivers, head contours indicate streams in the Mount Lofty Ranges are gaining (lines 5-6), yet on line 8, you state that "the infiltration capacity of the mountain block is limited". If the mountain block has such limited infiltration capacity, where is the water that causes the streams to be gaining coming from? If the streams are gaining, I assume it's because groundwater is flowing in, which would suggest the mountain block must be capable of receiving/transmitting water fairly well. The root of your 'low infiltration capacity' hypothesis seems to be on lines 6-7, where you observe that the principal exception to the 'streams are typically gaining in the Mount Lofty Ranges' rule is "along the upper reaches of rivers, i.e. where the stream order is small". A few thoughts on the upper reaches of streams:

Response: Agreed; the sentence was removed.

1. Are you sure the upper-reach sections of stream that you are discussing here are perennial? I have spent a lot of time in mountains, and a lot of that time has been spent sampling from springs. In my experience, virtually all perennial mountain streams in non-glaciated regions

originate as groundwater outflow. If, for example, you are relying on a GIS data set for stream locations/origination points, your data set may have sections of stream channel that are ephemeral digitized in (e.g., from air photos) with no distinction from perennial reaches of stream. By definition, an ephemeral stream channel will be reliant on overland flow/interflow after precipitation instead of groundwater. Have you ground-truthed any of these upper reaches of stream to make sure they are perennial? If the upper reaches that don't appear to be gaining are ephemeral, it removes support for your 'limited infiltration capacity' hypothesis (although as already stated, the fact that the rest of the stream reaches are gaining already goes against that hypothesis).

Response: Agreed; the upper reaches of streams are indeed ephemeral. We removed the statement about the low infiltration capacity of the mountain block.

2. Upper reaches of rivers in mountains tend to be in the highest-elevation areas, where wells (and thus head values) are typically the sparsest. Do you have the data density in the zones surrounding the upper reaches of rivers in the Mount Lofty Ranges to make the shapes/locations of your head contours definitive enough to truly tell whether the streams are gaining or losing? Looking at the head data points relative to stream headwaters in Figure 7, I'm not sure you have the data density to make this call. Also, looking at Figure 7, I'm not really able to see head contours 'veeing' (as Winter et al. (1998) show to indicate gaining or losing streams [depending on 'vee' direction relative to upstream/downstream]) to/from streams in most instances. If you have the data density to make plots as suggested in my comment on Section 3.3.1 and actually show the 'veeing', that would be much stronger support of your case.

Response: Indeed, the data density can be critical. However, in our study area the data density is not much different in the zones surrounding the upper reaches. We nevertheless added discussion on two issues related to data density. P10 L111-12: "A good compromise was found by setting this parameter to 1,200 m for the NAP focus area and to 800 m for the CAP focus area – reflecting a higher density of streams and data in the latter case.". P12 L19-20: "Note that one should not expect to see sharp "V" shapes where head contours cross streams (i.e., as in Figure 4) due to the limited data density.".

3. As mentioned in (2), upper reaches of rivers in mountains tend to be in the highest elevation areas, which means they would tend to be recharge zones, not discharge zones. Your Figure 1 suggests that mountaintops are recharge zones, and that it is necessary to get some distance downslope before you reach a groundwater discharge zone. This is a potential argument against the idea that "the infiltration capacity of the mountain block is limited". Recharge zones can have very high infiltration capacity, but because they are recharge zones, we don't expect discharge to occur. As a result, I don't believe that lack of discharge alone can be used as a test for low infiltration capacity. The upper stream reaches may be getting no groundwater inflow because they are in recharge zones instead of discharge zones.

Response: Agreed again; this statement was removed.

To sum up, I'm not sure the infiltration capacity of the mountain block is limited, but if you can present evidence more likely to convince me, I'd be willing to reconsider!

Response: We agree with this assessment, and implemented corrections accordingly, as highlighted in our previous responses.

p 12 L 7-8: see my earlier comment (p 10 L 5-8) that I think it is likely perennial streamflow in your system is generated by groundwater inflow, not by overland flow/interflow (also see your statement on line 19 of this page)

Response: We agree with the earlier comment; however, here we specifically refer to the periods of high flows, which would include the contribution from the ephemeral streams, i.e. overland flow and interflow.

p 13 L 11-13: see comment p 8 L 32-p 9 L 1; if pumping induces mixing between waters with different chloride concentrations, the chemical change should occur at the same rate as the hydraulic change

Response: As argued in our response to comment on p 8 L 32-p 9 L 1, we reiterate that the influence on hydraulic head would typically be seen earlier than the influence on chemistry, since the former refers to pressure travel time while the latter refers to solute travel time. This is the reason why, for instance, coastal wells can be operated for quite some time before observing seawater intrusion.

p 14 L 11: this could probably use some clarification—I think you are describing using an airlift pump at the same pressure in multiple wells and assigning higher T values to wells that yield a relatively high amount of water and lower T values to wells that yield a relatively low amount of water. Please note that many readers might be unfamiliar with airlift pumps, so this could merit a bit more explanation than you currently give.

Response: No; here we refer to air-lift yield testing conducted at the time of drilling, as described for instance in Williams et al. (2004). This was clarified and the latter reference was added (P15 L13-16): "Finally, Zulfic et al. (2010) found that bore yield based on air-lift testing conducted at the time of drilling (e.g., Williams et al., 2004) in the Mount Lofty Ranges did not increase beyond 100 m depth for most geology types. This finding can be interpreted as hydraulic conductivity being relatively low beyond that depth.".

*Reference*

Williams, L. J., Albertson, P. N., Tucker, D. D., and Painter, J. A.: Methods and hydrogeologic data from test drilling and geophysical logging surveys in the Lawrenceville, Georgia, area, Open-File Report 2004-1366, 2004.

p 11 L 19: what are you defining as "the front line"? Is it the fault that mostly runs along the eastern boundary of the NAP (but continues on through the CAP), is it the fault that mostly runs along the eastern boundary of the CAP, is it somewhere else? This should be clarified.

Response: What we meant is simply the "mountain front"; for clarity, this term is now used instead of "front line".

p 14 L 30-33: at the end of this sentence, you refer only to 'water found in streams running down the mountain' to contrast to water that recharges diffusively, but I think the important concept here is that [some/much of] the water flowing in the streams will become focused recharge in contrast to the diffuse recharge found outside stream zones. I think you need to make clear to the reader that the important concept here is diffuse vs. focused recharge.

Response: Accepted; the sentence has been reformulated to include this concept.

Conclusions: I will have an easier time agreeing with all the conclusions if you can address some of the earlier comments I made (e.g., if you can make figures as suggested in my comment on section 3.3.1, I can better believe that analysis of head contour shape adjacent to surface features allows you to distinguish gaining/losing reaches of streams).

Response: We thank you for the suggestions; the presentation should be more straightforward after revising as indicated above.

Grammar/spelling/etc. comments & comments on figures/tables:

Title: should read "electrical-conductivity data"

Response: No; "electrical conductivity" conventionally reads as one word, as do the terms "hydraulic head" and "hydraulic conductivity" for instances, which are never hyphenated.

p 2 L 16 (and numerous other instances): not sure of journal style, but typically "e.g." is followed by a comma, e.g., "e.g., Hely et al., 1971…"

Response: Accepted.

p 3 L 4: you make reference to a geographic location here, so you should 'call' a figure that shows that location. Your Figure 3 shows this area, but you need to move the call to this first mention of the area in the text (which will cause it to become Figure 2)

Response: Accepted.

p 3 L 23 (and any other instances): not sure of journal style, but typically "i.e." is followed by a comma, e.g., "i.e., the in-between-streams zones…"

Response: Accepted.

p 4 L 1: Winter et al., 1998 is cited here (and on line 8), but does not appear in the reference list, need to add a reference.

Response: Thank you for pointing out this omission; the reference was added to the list.

p 5 L 32: change "increasing" to "increase"

Response: Accepted.

p 7 L 27: change "aquifer test or drilling" to either "aquifer tests or drilling" or "aquifer testing or drilling"

Response: Accepted.

p 8 L 4: sentence is unclear—I'm not sure what is meant by the statement that the aquifer "was informed in the database for about two thirds of the wells".

Response: We meant (and changed for) "The name of the aquifer"; this should be clearer than just "The aquifer".

p 8 L 27: change "less restrictive filtering" to "less-restrictive filtering"

Response: Accepted.

Figure 6: two issues with this figure:

1. many of the contour labels are very difficult to read (too small); especially the red/pink labels for topography (and those are even worse over some of the areas with dark red-brown fill in the southeastern edge of the map).

Response: We understand the concern. In the new (zoomed-in) figures, the labels for topography are in grey.

2. you do not show a key for the contour fill colors that you use, if you provided one, that might help with problem (1)

Response: We took on the comment and removed the contour fill colours.

p 10 L 15: change "by" to "minus"

Response: Accepted.

Figure 10: the faults are often difficult to see (especially since one is nearly the same color as the CAP boundary (but a thinner line than the CAP boundary), and runs on/very near the CAP boundary for a good distance. Could the faults be some other color on this figure (the green of the Australian 'green and gold' color scheme might be one choice that would stand out a bit better)?

Response: Accepted; we changed the colour of the CAP boundary and made the fault lines thicker.

p 11 L 20: change "occur" to "occurs"

Response: Accepted.

p 11 L 25: cut the first instance of "salinity" on this line

Response: Accepted.

p 11 L 28: change "do not contribute either" to "also do not contribute"

Response: Accepted.

Figure 11/Table 1:

1. In Figure 11, you plot flow rate using the units ML/d, but in Table 1, you report mean flow rates in GL/y, which makes comparing the two difficult. Please pick one consistent set of units for flow rate/mean flow rate and use it in both Figure 11 and Table 1.

Response: Accepted: we now show all flow rates in ML/d.

2. At first glance, there appear to be two sets of 'paired' streams (e.g., the symbols used for South Para River and First Creek are identical to my eye, as are the symbols for Brownhill Creek and Gawler River). On closer examination, I can see a slight difference between the Grawler/Brownhill point colors, but it is tough to tell which is which (I think Brownhill is the lower-concentration data set on the plot); I can't make any difference out for the other 'pair'). Can you do something to make each data set more distinct? One suggestion: there are only so many colors you can use that work well unoutlined, but if your graphing software is able to outline points, they can become much more 'readable' (e.g., yellow circles on their own can be hard to see on a white background, but a yellow circle with a black border shows up well on a white background). Using outlined shapes might let you add a couple of distinct colors that would make it much easier to tell the data sets apart on the figure.

Response: Accepted: we improved the differentiation between the data of different streams by choosing more contrasted colours.

3. In Table 1, change two instances of "uS/cm" to "□S/cm"

Response: Accepted.

p 13 L 1: change "great" to something along the lines of "useful", "effective", etc.

Response: Accepted; we changed for "effective".

p 14 L 15: cut "based on"

Response: This would change the meaning; however we cut "used" earlier in the sentence to make it easier to read.

p 14 L 23: cut "only"

Response: Accepted.

p 14 L 25: add comma after 'contrast'

Response: Accepted.

p 14 L 28: cut "yet"

Response: Accepted.

p 15 L 1-2: change "this study proposes" to "we propose" [or if the journal style does not allow that, the more stilted "the authors propose"]. Your study can't propose anything, but you can!

Response: Accepted.

References:

Earman, S., Campbell, A. R., Newman, B. D., and Phillips, F. M., 2006. Isotopic exchange between snow and atmospheric water vapor: Estimation of the snowmelt component of groundwater recharge in the southwestern United States, Journal of Geophysical Research, 111: D09302, doi:10.1029/2005JD006470.

Simpson, E. S., Thorud, D. B., and Friedman, I., 1972. Distinguishing seasonal recharge to groundwater by deuterium analysis in southern Arizona. In: World Water Balance. Proceeding of the Reading Symposium, July 1970, Volume 3, International Association of Scientific Hydrology-UNESCO-WMO Studies and Reports in Hydrology, vol.11; Gentbrugge, Belgium, Publication No 94 of the International Association of Scientific Hydrology, pp 623–633

Winograd, I. J., Riggs, A. C.. and Coplen, T. B., 1998. The relative contributions of summer and cool-season precipitation to groundwater recharge, Spring Mountains, Nevada, USA, Hydrogeology Journal (6) 77– 93.

Winter T. C., Harvey, J. W., Franke, O. L., and Alley, W. M., 1998. Groundwater and Surface Water: A Single Resource, U.S. Geological Survey Circular 1139.

**Using hydraulic head, chloride and electrical conductivity data to distinguish between mountain-front and mountain-block recharge to basin aquifers**

Etienne Bresciani[1,2], Roger. H. Cranswick[1,3], Eddie W. Banks[1], Jordi Batlle-Aguilar[1,4], Peter G. Cook[1], Okke Batelaan[1]

[1]National Centre for Groundwater Research and Training, School of the Environment, Flinders University, Adelaide, SA 5001, Australia
[2]Korea Institute of Science and Technology, Seoul, 02792, Republic of Korea
[3]Department of Environment, Water and Natural Resources, Government of South Australia, Adelaide, SA 5000, Australia
[4]Kansas Geological Survey, University of Kansas, Lawrence, KS 66047, USA

*Correspondence to*: Etienne Bresciani (etienne.bresciani@flinders.edu.au)

**Abstract.** Numerous basin aquifers in arid and semi-arid regions of the world derive a significant portion of their recharge from adjacent mountains. Such rechargeRecharge can effectively occur through either stream infiltration in the mountain front zone (mountain-front recharge, MFR) or subsurface flow from the mountain (mountain-block recharge, MBR). While a thorough understanding of the recharge mechanisms is critical for conceptualizing and managing groundwater systemswater resource management, distinguishing between MFR and MBR is typically difficult. WeHere we present ana relatively simple approach that uses hydraulic head, chloride and electrical conductivity (EC) data to distinguish between MFR and MBR. These variables types of data are inexpensive to measure, and may bein many cases are readily available from hydrogeological databases in many cases. Hydraulic heads . In principle, hydraulic head can inform on groundwater flow directions and stream-aquifer interactions, while chloride concentrations and EC values can be used can help to distinguish between different water groundwater pathways if the sources if these have a distinct signature. Such information can provide evidence for the occurrence or absence of MFR and MBR. This concentrations. Electrical conductivity values can be converted to chloride concentrations using an empirical relationship, and hence can be used in a similar manner to chloride, thereby significantly increasing the data set. The practical feasibility and effectiveness of this approach isare tested through application tothe case study of the Adelaide Plains basin, South Australia. The recharge mechanisms of this basin have long been debated, in part due to difficulties in understanding the hydraulic role of faults. Both hydraulic head and chloride (equivalently, EC) data consistently, for which a wealth of historical groundwater level, chloride and electrical conductivity data is available. Hydraulic head data suggest that streams are gaining in the adjacent Mount Lofty Ranges and losing when entering the basin. Moreover, the data They also indicate that not only the Quaternary aquiferssediments but also the deeperunderlying Tertiary aquifers are recharged through MFR and not MBR. It is expected that this finding will have asediments receive significant impact on water resources managementrecharge from stream leakage in the region.mountain front zone. Chloride data also reveal clear spatial patterns suggesting that MFR dominates recharge of the low salinity

 This study demonstrates the relevance of using hydraulic head,  chloride and EC data  to distinguish between MFR and MBR.

**1 Introduction**

5   Numerous basin aquifers in arid and semi-arid regions receive a significant portion of their recharge from adjacent mountains, largely because the latter typically benefit from higher precipitation and lower evapotranspiration (Winograd et al., 1998; Wilson and Guan, 2004; Earman et al., 2006). Two recharge mechanisms can be recognized (Wahi et al., 2008): mountain-front recharge (MFR), which predominantly consists of stream infiltration in the mountain front zone, and mountain-block

10   recharge (MBR), which consists of subsurface flow from the mountain towards the basin. Here the mountain front zone is defined after Wilson and Guan (2004) as the upper zone of the basin,  between the basin floor and the mountain block (Figure 1a). The term MFR has traditionally been used to encompass the two recharge mechanisms described above, but it may be more appropriate to use it for the first one only.

15    Following Wahi et al. (2008), the collective process of MFR and MBR is referred to as mountain system recharge (MSR).

The distinction between MFR and MBR is important. The conceptualization of a basin groundwater system critically

20   depends on whether recharge occurs through MFR or MBR, as each of these mechanisms implies different groundwater flow paths, groundwater age and geochemical characteristics. MFR and MBR can also imply different responses to land and water resource management practices (both in the basin and the mountain) as well as to climate change. A good understanding of these mechanisms is thus essential for an effective coordinated management approach of water resources in basins and adjacent mountains

25    (e.g. Hely et al., 1971; Anderson, 1972; Maurer and Berger, 1997; Siade et al., 2015).

While various methods exist to estimate MSR as a bulk, characterizing the individual contributions of MFR and MBR is difficult. For instance, Darcy's law calculations and inverse groundwater flow modelling typically provide bulk MSR estimates (e.g., Hely et al., 1971; Anderson, 1972; Maurer and Berger, 1997; Siade et al., 2015). It is possible to consider

30   MFR and MBR independently in a groundwater flow model, but the solution to the inverse problem is more likely to be non-unique (e.g., Bresciani et al., 2015b). The water balance and chloride mass balance methods also provide bulk MSR estimates when the analysis is performed at the base of the mountain front zone or further downstream in the basin (e.g.,

Maxey and Eakin, 1949; Dettinger, 1989). Environmental tracers such as noble gases (e.g., Manning and Solomon, 2003), stable isotopes (e.g., Liu and Yamanaka, 2012) and radioactive isotopes (e.g., Plummer et al., 2004) can help to determine which of MFR or MBR is the dominant mechanism, but their analysis remains expensive and their interpretation can be difficult. The most robust approach for characterizing MFR and MBR might be the integrated analysis of all available hydraulic, temperature and concentration data through the coupled modelling of groundwater flow, heat and solute transport in the combined basin-mountain system (e.g., Manning and Solomon, 2005) – but it is also arguably the most complex approach.

. It is possible to consider MFR and MBR independently in a groundwater flow model, but the solution to the inverse problem is more likely to be non-unique (e.g. Bresciani et al., 2015b). The popular water balance and chloride mass balance methods also provide bulk MSR estimates when the measurements are made at the base of the mountain front zone or further downstream in the basin (e.g. Maxey and Eakin, 1949; Dettinger, 1989). Environmental tracers such as noble gases (e.g. Manning and Solomon, 2003), stable isotopes (e.g. Liu and Yamanaka, 2012) and radioactive isotopes (e.g. Plummer et al., 2004) can help to determine which of MFR or MBR is the dominant mechanism, but their analysis remains expensive and their interpretation can be difficult. The ultimate approach for characterizing MFR and MBR might be the integrated analysis of hydraulic, temperature and concentration data through the coupled modelling of groundwater flow, heat and solute transport in the combined basin-mountain system (e.g. Manning and Solomon, 2005) – but it is also arguably the most complex approach.

In this study, we explore alternatives to expensive and complex methods to investigate whether MSR to basin aquifers is dominated by MFR (Figure 1b) or MBR (Figure 1b) or MBR (Figure 1c), or if both types of recharge mechanismsprocesses are significant (Figure 1d). We focus on the use of hydraulic head, chloride ($Cl^-$)) and electrical conductivity (EC), which are inexpensive to measure and may bein many cases are readily available fromin large quantities in hydrogeological databases in many cases.. The general utility of hydraulic head and $Cl^-$ data to infer groundwater dynamics is well established (e.g., Domenico and Schwartz, 1997; Herczeg and Edmunds, 2000).(e.g. Domenico and Schwartz, 1997; Herczeg and Edmunds, 2000). Furthermore, EC values can be converted to $Cl^-$ concentrations (as demonstrated later), and hence can be used in a similar manner to $Cl^-$.. However, studies demonstrating the specific use of these data for the characterization of MSR mechanisms appear to be rare (Feth et al., 1966). This may reflect a traditionally low data density along mountain fronts, which are not generallytypically the prime locations for drilling groundwater wells due to an the often complex hydrogeology and expected lower yield than elsewhere in the basin (low aquifer thickness may typically be smaller, and prevailingyield (as these are the recharge conditions are not favourable to well yield).areas). However, with an ever-growing number of wells accompanying development of basin and mountain areas, data density and spatial coverage steadily increases, even in these zones.

After presenting a general rationale for the use of hydraulic head, $Cl^-$ and EC data to distinguish between MFR and MBR, theThe Adelaide Plains basin in South Australia (Figure 2) is used as a case study to test the relevance of the approach.. This semi-arid region features a typical sedimentary basin bounded by a mountain range – the Mount Lofty Ranges, from which

most of the recharge is believed to be  derived (Miles, 1952; Shepherd, 1975; Gerges, 1999; Bresciani et al., 2015a). Groundwater in this basin has been used for over a century for industry, water supply and agriculture. Nonetheless, and despite several recent studies, the relative contributions of MFR and MBR is still subject to debate. In particular,   the hydraulic role of faults (i.e., acting as barrier or conduit to flow) that run along the mountain front remains unclear (Green et al., 2010; Bresciani et al., 2015a; Batlle-Aguilar et al., 2017). As a result of the common use of groundwater, a relatively high density of wells exists in the region, including in the mountain front zone. Therefore, this case study provides a good example of the potential of the proposed approach.

**2 Rationale**

In this section, a generic rationale is presented for the use of  hydraulic head and Cl⁻ (or EC-derived Cl⁻) data to distinguish between MFR and MBR to basin aquifers. Hydraulic head and Cl⁻ data can be used independently, but as they are of different nature, it is expected that their simultaneous use will result in a more complete and reliable characterization of the recharge mechanisms.

**2.1 Using hydraulic head**

Hydraulic heads directly relate to groundwater dynamics. Consequently, hydraulic head patterns could theoretically enable the identification of groundwater flow paths, both in mountains and basins. Specifically, four types of analysis are suggested  that could inform the likely occurrence or absence of MFR and MBR:

1. Assessment of the correlation between hydraulic head and topography. In the mountain block, a good correlation would suggest that groundwater flow is dominated by local flow systems as opposed to regional flow systems (Tóth, 1963). This would imply that only a small portion of the recharge occurring over the mountain would make its way towards the basin. In fact, in this case MBR would be mostly limited to the recharge occurring over triangular facets in between stream catchments at the base of the mountain block (Figure 3). Here, the recharge is less likely to be routed towards mountain streams, and instead it may be  routed towards the basin  (Welch and Allen, 2012). In  the mountain front zone,  a good correlation between hydraulic head and topography would suggest that groundwater discharges to streams, so that MFR from stream leakage would be limited or non-existent.

2. Analysis of the shape of head contours adjacent to surface water features to identify losing and gaining stream conditions. It is well known that head contours show a curvature pointing in the downstream direction where the contour lines cross a losing stream (due to the mounding induced by groundwater recharge) (Figure 4a), whereas they show a curvature pointing in the upstream direction where the contour lines cross a gaining stream (due to the depression induced by groundwater discharge) (Figure 4b) (Winter et al., 1998).

Performing such analysis in the mountain block should indicate whether mountain groundwater appears mostly routed towards local streams, which would make it less likely for MBR to be significant. Additionally, performing such analysis in the mountain front zone should allow for testing the occurrence or absence of MFR (at least in the form of stream infiltration, which is the predominant form of MFR (Wilson and Guan, 2004)).).

5    3. Comparison of stream levels with nearby groundwater levels. A; a stream level higher than nearby groundwater levels would indicate a potential for groundwater discharge to stream infiltration, while the opposite would indicate a potential for groundwater discharge to stream.stream infiltration (e.g. Winter et al., 1998). If the data density is low, this analysis may be preferable over the previous one (#2) as it does not require head contours to be accurately determined. However, it can only inform on a *potential* interaction: groundwater discharge or recharge would be
10   significant only if the hydraulic conductivity of the streambed is high enough. In contrast, the previous analysis (#2) could give a more definite answer, because the curvature of head contours at some distance from the stream should only be visible if the groundwater-surface water interaction is significant relative to other flow components (i.e., horizontal flow).

     4. Evaluation of the vertical head gradient in the mountain front zone. Recharge; recharge areas are associated with a
15   decrease of hydraulic head with depth, while discharge areas are associated with an increase of hydraulic head with depth (e.g., Wang et al., 2015)(e.g. Wang et al., 2015). Hence, in the mountain front zone, a head decrease with depth would suggest that MFR occurs (at a rate that depends on the vertical hydraulic conductivity of the aquifer). In), while in contrast, an absence of head decrease (or a head increase) with depth would suggest that MFR does not occur.

20   In cases where faults run between the basin and the mountain, it may be tempting to study the difference inof hydraulic head between the two sides of the fault zones. Intuitively., with the idea that a large head difference would indicate that a fault zone constitutes a hydraulic barrier to flow in the direction perpendicular to it (e.g., Bense et al., 2013), and consequently that MBR would be low. However, a large head difference across a fault zone may not always imply that the fault zone constitutes a hydraulic barrier.(e.g. Bense et al., 2013), and consequently that MBR would be low. However, a large
25   difference in hydraulic head across a fault zone may not always imply that the fault zone constitutes a hydraulic barrier, as discussed in the following. Let us consider the hypothetical case of a sedimentary layer overlying a basement of relatively low hydraulic conductivity, and thatwhich features a sharp transition in elevation as a consequence of faulting (Figure 2a).Figure 2a). The hydraulic conductivity of the fault zone itself is notassumed to be no different fromto that of the embedding materials (i.e., the fault only implies a difference in basement elevation)., In this simple configuration, it happens
30   that if the groundwater level right below the fault (as a result of downstream controls) is lower than the elevation of the basement elevation above of the fault – as a result of downstream hydraulic controls, the groundwater level above the fault isbecomes essentially 'disconnected' from the lower part (Figure 2b). This issystem because in all cases, the groundwater level above the faultit has to satisfy a minimum height (i.e., transmissivity) for groundwater to flow there. (Figure 2b). Hence, inthis example demonstrates thatease, a large difference in head can exist across the fault zone despite the fact that

the fault zone itself has no specific (low) hydraulic conductivity, It should also be noted that regardless of the cause, the implications of a large difference in head in terms of the amount of flow eventually crossing the fault zone is far from obvious, as it depends on the hydraulic conductivity of  the fault or the basement (which is in either case difficult to determine). A more relevant analysis may.  be  to investigate whether or not the  hydraulic head above of the fault will be so high (relative to topography) as to imply local groundwater discharge to mountain streams instead of  lateral flow across the fault towards the basin. In other words, what matters is the partitioning of the mountain groundwater between these two pathways. This is precisely what the first three types of analysis presented above should contribute to determine.

**2.2 Using chloride**

Chloride ($Cl^-$) is a naturally occurring ion in groundwater that is generally considered as conservative in geochemical studies (Clark and Fritz, 1997). If there is no removal or addition of $Cl^-$ in the aquifer, and if the effects of dispersion (i.e., mixing of water from different flow paths) can be neglected, the $Cl^-$ concentration will be constant along each groundwater flow path (Bresciani et al., 2014). Under such conditions, if the potential MFR and MBR sources have a different $Cl^-$ concentration, $Cl^-$ could be an excellent tracer to distinguish between these two recharge mechanisms.

In many cases, $Cl^-$ in groundwater primarily originates from atmospheric deposition (Allison et al., 1994). The rate of atmospheric deposition depends on a number of factors including distance to the source (oceanic or terrestrial), elevation, terrain aspect, slope, vegetation cover and climatic conditions (Hutton and Leslie, 1958; Guan et al., 2010b; Bresciani et al., 2014). Other potential sources of $Cl^-$ in groundwater include anthropogenic inputs (e.g., salting of roads, irrigation, application of fertilizer, leakage from septic/sewage systems) and dissolution of Cl-bearing minerals. $Cl^-$ removal from solution is unlikely as Cl does not easily adsorb onto clays or precipitate as mineral (Clark and Fritz, 1997). The $Cl^-$ concentration in recharge water also depends on evapotranspiration, which leaves $Cl^-$ in solution, implying its enrichment (Eriksson and Khunakasem, 1969; Allison et al., 1994) (a growing vegetation could in theory counter this effect since Cl is a nutrient for plants (White and Broadley, 2001), but in practice the uptake of Cl from soil by most plant species is insignificant (Allison et al., 1994)). Thus, depending on how variable the above controlling factors are, the $Cl^-$ concentration in mountain groundwater – i.e., the potential MBR source – may show significant spatial and temporal variability. On the other hand, the $Cl^-$ concentration in stream water entering the basin – i.e., the potential MFR source – strongly depends on the streamflow generation mechanisms. If the mountain streams are supported by large proportions of overland flow or interflow, the $Cl^-$ concentration in stream water entering the basin will tend to be lower than that in mountain groundwater, because these streamflow generation mechanisms imply relatively little evapotranspiration. In contrast, if the mountain streams are mostly supported by groundwater discharge, the $Cl^-$ concentration in stream water entering the basin will tend to have an integrated value of the mountain groundwater $Cl^-$ concentration. In conclusion, while no general statement can be made, chances are that the potential MFR and MBR sources have a distinct $Cl^-$ signature.

Chloride (Cl) is a naturally occurring element in groundwater that is relatively non-reactive compared to other elements. This makes it a good conservative tracer of groundwater, except in particular cases where lithology can be an important source of Cl (e.g. Claassen and Halm, 1996). Thus, in many environments, the Cl concentration can be assumed to remain equal to that of recharge along the groundwater flowpaths (if the effects of dispersion can be neglected (e.g. Bresciani et al., 2014)).

5   Therefore, if the Cl concentration of the potential MFR source has a distinct signature from that of the potential MBR source, it could provide an excellent tool to distinguish between these two recharge mechanisms.

Cl in groundwater originates from atmospheric deposition, of which the rate depends on a number of factors including distance to the source (oceanic or terrestrial), elevation, terrain aspect, slope, vegetation cover and climatic conditions (Hutton and Leslie, 1958; Guan et al., 2010; Bresciani et al., 2014). Groundwater Cl concentrations also depend on

10   evapotranspiration, which leaves Cl in solution, implying its enrichment (Eriksson and Khunakasem, 1969), and on the spatial redistribution of recharge through groundwater flow. Hence, groundwater Cl concentration in mountains can be expected to show significant spatio-temporal variability. If MBR occurs, the associated Cl concentration depends on where (and when) the water initially originates — i.e. the infiltration point. On the other hand, if MFR occurs, the associated Cl concentration depends on streamflow generation mechanisms, i.e. overland flow, interflow and groundwater discharge. In

15   particular, mountain streams are often supported in significant proportions by overland flow or interflow, in which case they could have a lower Cl concentration than mountain groundwater since these mechanisms imply relatively little evaporation. Therefore, potential MFR water and potential MBR water are likely to have distinct Cl signatures.

In this study, the proposed strategy consists of analysing three types of water for Cl⁻: groundwater in the basin, stream water at the mountain front, and groundwater in the basin, groundwater in the mountain, and stream water in the mountain front

20   zone.mountain block near the basin. Assuming steady-state concentrations and conservative Cl⁻, groundwater in the basin should have the same concentration as stream water inat the mountain front zone in the case of if it comes from MFR (further assuming that transpiration from plants after stream infiltration and potential mixing with diffuse recharge are negligible), while it ). In contrast, the basin groundwater should have the same concentration as the mountain groundwater if it comes from MBR. For the latter to be assessed properly, it is important to assess groundwater Cl concentration in the

25   mountain in the case of MBR. As muchas close as possible, the analysis should focus on comparing points that are not too far apart from another and along presumed flow paths, such as to the basin to reduce potential risks of misinterpretation caused by spatiotemporalthe spatial variability inof Cl⁻ concentration and dispersion. In particular, MBR should be most reliably assessed by comparing the groundwater Cl⁻ concentration in the uppermost part of the basin to that in the lowermost part of the mountain, along lines running perpendicular to the mountain front.

30   Electrical conductivity (EC) values areis known to be strongly correlated to Cl⁻ concentrations (Guan et al., 2010a). Therefore, EC data, and hence can be converted to Cl⁻ data if a relationship between the two can be assumed. Typically, EC is more routinely measured than Cl⁻, and thus this should significantly increase the dataset. Ideally, an empirical relationship between EC and Cl⁻ should be developed based on available pair measurements in the study area. As EC is typically more routinely measured than Cl, this should significantly increasing the data set.

**3 Case study**

**3.1 Study area and background**

The Adelaide Plains (AP) basin is a coastal sedimentary embayment of 1,700 km$^2$ in South Australia (Figure 2)
The area is bounded by the Mount Lofty Ranges to the east and south, by the Light River to the north, and by the Gulf Saint
Vincent to the west. It can be split into two sub-basins: the Central Adelaide Plains (CAP) sub-basin south of Dry Creek, and
the Northern Adelaide Plains (NAP) sub-basin north of Dry Creek. The topographic gradient is more pronounced in the CAP
and adjacent mountains (regional slopes of about 0.8 % and 7 %, respectively) than in the NAP and adjacent mountains
(regional slopes of about 0.3 % and 2.5 %, respectively). Torrens River and Gawler River are the largest rivers in the
CAP and in the NAP, respectively. A number of streams run down from the Mount Lofty Ranges, either feeding these
rivers or flowing directly into the ocean.

Precipitation is relatively low and potential evapotranspiration is high in this semi-arid area. The average
rainfall is 445 mm yr$^{-1}$ (no snowfall) and the annual average maximum daily temperature of 21.6 °C at Adelaide Airport
station located near the coast (station number 23034, 1970–2013; Australian Government, Bureau of Meteorology)~~.), which
is located near the coast., and instead~~
most of the recharge is believed to be derived from the adjacent Mount Lofty Ranges. The latter receive an average rainfall
of 983 mm yr$^{-1}$ and negligible snowfall (i.e., only exceptionally and in insignificant quantities) at Mount Lofty Cleland
Conservation Park (station number 23810, 1970–2013; Australian Government, Bureau of Meteorology), i.e. more than
twice than, and~~ experiences cooler temperatures with an annual average maximum daily temperature
of 15.2 °C at Mount Lofty (station number 23842, 1993–2007; Australian Government, Bureau of Meteorology). The
majority (77 %) of the rainfall in the Mount Lofty Ranges occurs during autumn and winter (May–September) (station
number 23810, 1970–2013), suggesting a strong seasonality of the recharge.

The basin comprises complex, spatially-dependent sequences of Quaternary and Tertiary sedimentary deposits. ~~The
Quaternary sediments are dominated by fluvio-lacustrine clay interbedded with sand and gravel, while the Tertiary sediments
are dominated by sand, sandstone, limestone, chert, marl and shell remains interbedded with clay~~ (Gerges, 1999). The
Quaternary sediments are dominated by fluvio-lacustrine clay interbedded with sand and gravel. The Tertiary sediments are
dominated by sand, sandstone, limestone, chert, marl and shell remains interbedded with clay. A number of faults dissect the
basin. Among these, the Eden-Burnside Fault and the Para Fault are of
primary interest in this study since they run along the foothill, almost at the margin of the CAP and the NAP sub-basins,
respectively (Figure 2) The total thickness of the sedimentary units increases sharply on the downthrown side of
the major faults (up to 400 m in places). The thickness of the Quaternary sediments ranges from 0 to about 140 m
across the basin (Figure 4a), while that of the Tertiary sediments ranges from 0 to about 500 m (Figure 4b). The Tertiary
sediments are directly outcropping in the northeast part of the CAP. The basement of the basin and the Mount Lofty Ranges
are mostly comprised of Proterozoic fractured rocks of various lithologies including slate, phyllite, quartzite, limestone and

dolomite. Superficial sedimentary deposits (typically less than 20 m in thickness) also exist locally in the Mount Lofty Ranges.

Up to six semi-confined aquifers (named Q1 to Q6) are recognized withinin the Quaternary sediments from the central to western side of the basin (Gerges, 1999) (i.e., downstream. west of the mountain front zone). These aquifers contain water of variable salinity with a median value of around 1,300 mg L$^{-1}$. The underlying Tertiary sediments are generally subdivided into four aquifers (named T1 to T4) over a large part of the basin. However, there is no clear hydrogeological distinction between the various Tertiary sediments along most of the mountain front zone in both sub-basins, and thus in this area they are considered to form a single undifferentiated Tertiary aquifer (Gerges, 1999; Zulfic et al., 2008; Baird, 2010). Simplified cross-sections of the aquifers in the NAP and CAP sub-basins are shown in Figure 7a and Figure 7b, respectively. Salinity is relatively low in the upper aquifer (T1) with a median value of around 600 mg L$^{-1}$, slightlyand is higher in the T2 aquifer with median values of around 1,000 mg L$^{-1}$, and significantly higher in deeper aquifers with median values of around 1,000 mg L$^{-1}$, 8,400 mg L$^{-1}$ and 40,000 mg L$^{-1}$ in T2, T3 and T4, respectively (but note however that very few data are available from the latter twoT3 and T4 aquifers). Because they present large areas of good salinity and yield, the T1 and T2 aquifers have been used since 1914 for occasional water supply, irrigation and industrial activities, and are currently the main targets of groundwater extraction in the AP (Gerges, 1999; Zulfic et al., 2008). Long-termPermanent, large cones of depression in both of these aquifers and forecasted increases in groundwater demand raise concerns about the sustainability of extraction in the coming years (Bresciani et al., 2015a). Risks are related to both potential depletion of the resource and rise in salinity from the migration of higher-salinity groundwater, which could make groundwater unusable. To better assessestimate these risks, a thorough understanding of the recharge mechanisms to these aquifers is necessary.

Early investigations suggested that the natural (i.e., pre-development) recharge to the Tertiary aquifers of the basin was dominated by stream infiltration along the mountain front (i.e., MFR) (Miles, 1952; Shepherd, 1975).(Miles, 1952; Shepherd, 1975). In contrast, subsequent investigations suggested that the natural recharge of the Tertiary aquifers was dominated by subsurface flow from the Mount Lofty Ranges (i.e., MBR) (Gerges, 1999, 2006). The latter conceptual model has formed the basis of most investigations of the Tertiary aquifers since its presentation, and underpinned the development of a number of groundwater flow and transport models of the basin aquifers (Jeuken, 2006a, b; Zulfic et al., 2008; Georgiou et al., 2011; Bresciani et al., 2015b). However, studies from Green et al. (2010) and Bresciani et al. (2015a) produced results supporting the hypothesis that both MFR and MBR could beoccur in significant proportions. To further investigate this question, the present study provides a re-appraisal of available hydraulic head, Cl$^-$ and EC data through application of the above rationale described above.

**3.2 Datasets**

**3.2.1 Hydraulic head dataset**

Hydraulic head data in the AP catchment (i.e., the area including both the basin and contributing mountain areas based on surface topography) were retrieved from the WaterConnect database (www.waterconnect.sa.gov.au, Government of South
5 Australia) on 04/11/2016. The collection dates span more than a century, the earliest measurements being from 1906 and the latest from 2016. The data were filtered out for unsuitable measurements such as measurements taken during pumping, aquifer testing or drilling. After filtering, 111,538 hydraulic head measurements from 9,561 wells were obtained.

The data were subsequently split according to three aquifer groups: the AP Quaternary aquifers, the AP Tertiary aquifers ('AP' in these expressions will be omitted in the remaining text) and the Mount Lofty Ranges aquifers.
10  This grouping is relevant in view of the hydrogeological characteristics of the system and the objective of the study. In particular, we did not distinguish between the T1 and T2 aquifers (i.e., the two main aquifers of the AP basin) because, as mentioned earlier, they are undifferentiated along most of the mountain front. In  the Mount Lofty Ranges, the presence of complex fracture networks and high relief can induce the blurring of otherwise depth-dependent hydraulic signals, and so splitting the data according to depth in this area may not be
15 very meaningful, while it would  reduce data density.

The name of the aquifer into which the wells were screened was informed in the database for about two thirds of the wells (6,209). This allowed for assignment of these wells to one of the above aquifer groups. For the remaining one third of wells, the aquifer group for the wells located in the basin was determined by comparing the well mid-screen elevation to the bottom elevation of the Quaternary sediments and to the top elevation of the basement.
20  The largest number of wells was from the Quaternary aquifers (3,964), followed by the Mount Lofty Ranges aquifers (3,589) and the Tertiary aquifers (1,768). Wells screened into the basement of the basin were disregarded (240 wells).

Groundwater level fluctuations can be an issue for data interpretation. In particular, as this study focuses on natural recharge mechanisms, the impact of pumping constitutes a potentially important bias. It should be noted that the density of hydraulic
25 head data is higher in areas of lower  groundwater salinity, which coincides with areas that have experienced greater changes due to pumping. The measurements made before the main development period (i.e., before 1950) may have been less affected by pumping than more recent measurements, but limiting the analysis only to these measurements would dramatically reduce the data density. In addition, even the earliest measurements may not be free of pumping influence, since it is likely that these were precisely taken to monitor the impact of pumping. Hence, instead of subjectively fixing an
30 arbitrary date beyond which the data would be excluded, all  data were retained regardless of the measurement date. For each of the wells that had multiple measurements, the temporal mean hydraulic head was calculated in an effort to smooth out the measurement errors and temporal fluctuations. The analysis focuses on these mean values,

**3.2.2 Chloride datasetdata set**

Groundwater Cl⁻ data in the AP catchment were also retrieved from the WaterConnect database on 04/11/2016. This datasetThe Cl data set was extended using the more commonly available EC data from the database. A strongEC is known to be strongly correlated to Cl, and a relationship between EC and Cl⁻ was found from thus derived using 1,559 pair

5    measurements (R² = 0.9996, Figure 8). In wells where only(Figure 5). All EC data were available, EC values were subsequently converted into Cl⁻ concentrations data using this robust relationship. (R² = 0.9996). In total, 34,145 Cl⁻ or EC-converted Cl⁻ data (simply referred to as Cl⁻ data in the following) from 12,660 wells were obtained (i.e., slightly more than for hydraulic heads, partly due to a less-restrictive filtering, i.e., keeping measurements taken during pumping or aquifer testing).). The collection dates span the same period as for the hydraulic head data.

10   The Cl⁻ data were subsequently split according to the same three aquifer groups were distinguished as indicate above for the hydraulic head data. The, and the same procedure was also applied to determine the aquifer group tointo which the wells belongare screened. The largest number of wells was from the Quaternary aquifers (4,963), followed by the Mount Lofty Ranges aquifers (4,395) and the Tertiary aquifers (2,963).

Pumping may also have impacted Cl⁻ concentrations by inducing migration of the original groundwater, whose concentration

15   is spatially variable (although this effect may be seen later than on hydraulic heads since solutes travel times are typically longer than pressure travel times). Also. Namely, as for hydraulic head data, the density of Cl⁻ data is higher in areas that have experienced pumping. Additionally, in the uppermost aquifers, irrigation may have locally influenced the However, the impact of pumping on Cl⁻ concentration. Nonetheless, following the same reasoning concentrations is expected to be less important than on hydraulic heads because groundwater chemistry typically responds less rapidly to perturbations than

20   groundwater hydraulics. Hence, as for hydraulic heads, all available Cl⁻ data were retained regardless of the measurement date.. For each of the wells that had multiple measurements, the temporal mean Cl⁻ concentration was calculated. The and was used in the analysis focuses on these values.

Flow rate and EC data from a number of streams running down from the Mount Lofty Ranges into the AP basin were also retrieved from the WaterConnect database. Six gauging stations were located close enough to the mountain front zone to be

25   relevant to the current study. Details on this datasetdata set are given in Table 1. The reported EC values of surface water were converted into Cl⁻ concentrations using the same relationship as developed for groundwater. This approach, which is deemed appropriate given the common origin of these waters, even though potentially different chemical reactions might slightly affect the relationship.

**3.3 Data analysis**

30   Given the relatively large area investigated, the analysis presented below concentrates on two "focus areas" that cover the transition between the Mount Lofty Ranges and the AP basin: one at the margin of the NAP sub-basin and one at the margin

of the CAP sub-basin (locations indicated in Figure 2). Figures for the entire study area are also available in Supplementary Material (Figures S1-S12). These do not call for a different interpretation.

**3.3.1 Hydraulic heads**

Hydraulic head maps were constructed for  the three aquifer groups (Quaternary aquifers,
5   Tertiary aquifers and Mount Lofty Ranges aquifers) (Figure 6 and Figure 10). In constructing these maps, the  choice of the interpolation method and associated parameters revealed to be critical. The classical Inverse Distance Weighting method would produce the well-known 'bull's eye' effect around individual data points. This could severely compromise the interpretation of head contours. Instead, the Diffusion Kernel interpolation method from the Geostatistical Analyst extension of ArcGIS 10.4.1 was used. This method allows for a more
10  realistic interpolation when the underlying phenomenon governing the data is diffusive, as is the case for hydraulic heads. The most important parameter in this method is the bandwidth, which is used to specify the maximum distance within which data points are used for prediction. Taking this parameter too small would undermine the prediction capability as many areas would remain uncovered by the interpolation, while taking it too large would produce overly- smoothed results. A good compromise was found by setting this parameter  to 1,200 m
15   for the NAP focus area and to 800 m for the CAP focus area – reflecting a higher density of streams and data in the latter case. Topographic contours were also constructed. To facilitate the comparison with head contours, these were created after application of a circular moving-average window to the topography using a radius that matches the bandwidth used in the  interpolation method for hydraulic head (i.e., 1,200 m in the NAP focus area and 800 m in the CAP focus area).
20  Figure 6a displays hydraulic head and topographic contours in the NAP focus area, showing the Quaternary aquifers on the basin side and the Mount Lofty Ranges aquifers on the mountain side. The results are quite contrasted between the mountain and the basin. In the mountain, head contours follow the topographic contours relatively closely, and their shape is most often indicative of gaining stream conditions. Note that one should not expect to see sharp "V" shapes where head contours cross streams (i.e., as in Figure 4) due to the limited data density. Instead, head contours are smoothly curved. In the basin,
25  head contours do not closely follow the topographic contours, and their shape is generally indicative of losing stream conditions (especially close to the basin margin). Figure 9b also displays hydraulic head and topographic contours in the NAP focus area, but showing the Tertiary aquifers on the basin side instead of the Quaternary aquifers. Head contours in the Tertiary aquifers are generally indicative of focused recharge along streams, but at a somewhat larger-scale, i.e., showing wider curvatures than in the Quaternary aquifers (mostly around Gawler River and Little Para River). Figure 9c and a
30  ~~displays head contours in the Quaternary aquifers and the Mount Lofty Ranges aquifers, while Figure 6b displays head contours in the Tertiary aquifers and the Mount Lofty Ranges aquifers. Only the most relevant area of the catchment is shown for a better visualization. The colours and contour interval are different in the basin and in the Mount Lofty Ranges to accommodate the fact that the range of head variations is much larger in the mountain than in the basin. Topographic~~

contours are shown with the same interval as for hydraulic head contours (i.e. different in the basin and the mountain). The topographic contours were calculated after application of a circular moving-average window of 1,200 m radius to the topographic map (i.e. matching the bandwidth used in the interpolation method for hydraulic head) to facilitate comparison with the hydraulic head contours. The figures reveal that the shape of hydraulic head contours and topographic contours is quite similar in the Mount Lofty Ranges aquifers, indicating a good correlation between hydraulic head and topography. This suggests, at least qualitatively, that groundwater flow is dominated by local flow systems in the Mount Lofty Ranges, and by consequence that the source of MBR may be limited to the recharge occurring over 'triangular facets' at the base of the mountain (see section 2.1). In contrast, head contours do not appear to follow a subdued expression of topographic contours in the mountain front zone, both in the Quaternary and Tertiary aquifers. This suggests that the streams are at least not gaining, and thus potentially losing.

Figure 7a and Figure 7b are essentially the same as Figure 6d display analogous results for the CAP focus area. Similarly to above, in the mountain, head contours are relatively well correlated with topographic contours and their shape is generally indicative of gaining conditions. In the basin, head contours in the Quaternary aquifers are not very distinct from topographic contours, but nevertheless tend to indicate losing rather than gaining stream conditions close to the basin margin (a and Figure 6c). In the Tertiary aquifers, head contours are quite distinct from topographic contours and are quite clearly indicative of focused recharge along a majority of streams (Figure 9d). Near Glen Osmond Creek and Brownhill Creek, groundwater flow predominantly appears oriented towards the southwest, which may result from the bedrock sloping in this direction (the Tertiary sediments thickness can be seen to increase in Figure 6b).

Figure 10a-d display the difference, in every point, between river head (approximated by the topographic elevation of the nearest river) and groundwater head. Figure 10a shows the NAP focus area, with the Quaternary aquifers on the basin side and the Mount Lofty Ranges aquifers on the mountain side. In the mountain, the results generally reveal a potential for gaining stream conditions along large portions of the main rivers (i.e., North Para River, South Para River and Little Para River). Potential losing stream conditions are indicated around the upper reaches of streams, suggesting that these are not supported by groundwater discharge, but are rather initiated by overland flow or interflow. This observation is consistent with the fact that most of the stream headwaters in the Mount Lofty Ranges are ephemeral. Potential losing stream conditions are also observed locally around a few streams in the lowest part of the Mount Lofty Ranges (e.g., South Para River and Smith Creek). This observation is not in line with the interpretation of head contours made from Figure 9a. This inconsistency might be an artefact of the temporal averaging of hydraulic heads, i.e., the hydraulic heads might be on average lower than the river head but the stream might still be gaining due to important groundwater discharge in some periods (but this explanation remains a hypothesis). In the basin, the Quaternary aquifers are revealed as potentially receiving water from streams everywhere, and especially close to the basin margin where the head difference is the largest. Figure 10b shows the head difference between the Quaternary aquifers and the Tertiary aquifers on the basin side, such as to investigate the vertical connection between these aquifers (on the mountain side, this figure is identical to Figure 10a). The hydraulic head appears larger in the Quaternary aquifers than in the Tertiary aquifers over most of the area. This indicates a

potential for downward groundwater leakage from the Quaternary aquifers to the Tertiary aquifers. The rate at which this leakage occurs is nonetheless difficult to estimate, since it is also function of the effective vertical hydraulic conductivity and vertical distance between these units, which are largely unknown. Similar observations and interpretations can be made of the CAP focus area (Figure 10c and Figure 10d).

5    Most observations from Figure 9 and Figure 10 suggest that groundwater flow is dominated by local flow systems in the Mount Lofty Ranges. This indicates that only a small proportion of the recharge occurring over the mountain may make its way towards the basin. Hence, if MBR occurs, it would be probably limited to the routing of the recharge occurring over triangular facets in between stream catchments at the base of the mountain (see section 2.1). By contrast, the results suggest that MFR is an important recharge mechanism for both the Quaternary aquifers and the Tertiary aquifers of the AP basin.

10   b, respectively, but rivers are shown and not topographic contours. Different sets of figures appeared necessary to improve the readability and allow for a more focused interpretation. The shape of head contours near streams is generally indicative of gaining conditions in the Mount Lofty Ranges. Exceptions are principally located along the upper reaches of rivers, i.e. where the stream order is small. The latter observation suggests that streams are not primarily initiated by groundwater discharge but by overland flow or interflow, and, as a consequence, that the infiltration capacity of the mountain block is
15   limited. In contrast, in the mountain front the shape of head contours near streams is in most instances indicative of losing conditions. A striking symmetry is even observed for some of the main rivers entering the basin, with head contours pointing upstream at the base of the Mount Lofty Ranges while pointing downstream in the mountain front zone, indicating a sudden change of conditions from gaining to losing (e.g. along the Gawler River and its tributaries). Remarkably, indications of losing river conditions are observed not only in the Quaternary aquifers but also in the Tertiary aquifers, suggesting that
20   significant amounts of water losses to the Quaternary aquifers reach the underlying Tertiary aquifers.
Figure 8a and Figure 8b display the result of the subtraction, at every point, of the nearest river elevation by the hydraulic head. The first figure shows the result for the Quaternary aquifers and the Mount Lofty Ranges aquifers, while the second figure shows the result for the Tertiary aquifers and the Mount Lofty Ranges aquifers. In the Mount Lofty Ranges, these figures corroborate the interpretations made above regarding the groundwater-surface water interactions: most rivers appear
25   to be potentially gaining (as seen from the blueish-coloured areas, which indicate a potential for groundwater to flow towards the nearest river), except in their upper reaches (as seen from the reddish-coloured areas, which indicate a potential for groundwater to receive water from the nearest river). The Quaternary aquifers are revealed as potentially receiving water from rivers over the entire basin, and especially in the mountain front zone, where the difference between nearest river elevation and hydraulic head is the largest. The Tertiary aquifers globally show the same patterns, except over a few small
30   areas near the mountain front in the CAP sub-basin, where a potential for groundwater to flow towards streams is indicated – namely around a portion of Torrens River. The fact that no area shows up as potentially gaining in the western part (i.e. the lower part) of the basin can be surprising, as one may expect to find groundwater discharge areas here, particularly near the coast. Under pre-development conditions, the hydraulic head in these areas was indeed higher than the land surface (Gerges,

1999). There is no doubt that this observation reveals the effect of pumping, which is known to be especially intense in the western part of the basin in both the T1 and T2 aquifers (e.g. Bresciani et al., 2015a).

The vertical head gradient in the basin was investigated through the head difference between the Quaternary and Tertiary aquifers. The results show that the most of the mountain front zone is characterized by a significant downward head gradient, with up to 59 m head difference (Figure 9). This indicates a downward leakage of groundwater from the Quaternary to the Tertiary aquifers. The rate at which this leakage occurs is of course also function of the effective vertical hydraulic conductivity and relevant distance between these units, which are largely unknown. Note that in Figure 9 the large red zone located towards the centre of the NAP near the Gawler River reflects the impact of extensive historical and ongoing groundwater extraction from the T2 aquifer.

**3.3.2 Chloride concentrations**

A Cl⁻ concentration maps were also map was constructed for each of the three aquifer groups (Quaternary aquifers, Tertiary aquifers and Mount Lofty Ranges aquifers) (). Here using the Inverse Distance Weighting interpolation method from the Geostatistical Analyst extension of ArcGIS 10.4.1 was used. This method is appropriate for the Cl⁻ concentrations values because their analysis does not especially make use of the shape of concentration focus is not on the contours, and hence the 'bull's eye' effect is not really an issue. Furthermore, Cl⁻ cannot be assumed to does not result from a diffusive process since at regional scale (advection typically dominates at the scale of this study, scale), and so the Diffusive Kernel method would be inappropriate. The Inverse Distance Weighting interpolation method also has the advantage of being exact at the data points. The power parameter was set to 2, and a standard neighbourhood was used with 15 maximum neighbours and 10 minimum neighbours. The same parameters were used for both NAP and CAP focus areas.

Figure 10a shows the Cl⁻ concentrations in the Quaternary aquifers and the Mount Lofty Ranges aquifers, while Figure 10b shows the Cl concentrations in the NAP focus area. This figure reveals Tertiary aquifers and the Mount Lofty Ranges aquifers. The figures reveal a strong correlation between stream locations and low Cl⁻ concentration zones and the location of the main rivers on the in the basin side (Gawler River and Little Para River)., both in the Quaternary aquifers and Tertiary aquifers. It seems highly unlikely that such a correlation would be observed if MBR was the main recharge mechanism. By contrast Furthermore, no obvious such correlation can be found seen in the mountain Mount Lofty Ranges aquifers. Here, the Cl⁻ concentration mostly appears correlated with elevation, with lower values occurring at higher elevations. This trend is expected, since the rate of evapotranspiration – which largely controls Cl⁻ concentration – is expected to decrease with elevation as a result of higher rainfall and lower temperature. In line with these observations, there is a clear discontinuity in Cl⁻ concentration at the transition between the mountain and the basin, almost everywhere along the mountain front line. This suggests that little or no hydraulic connection occurs occur between the mountain and the basin through the subsurface (i.e., MBR is probably insignificant). Similar observations hold in b, where . In particular, the Tertiary aquifers are shown lowest concentrations found along streams in the basin instead the Quaternary aquifers. The zones of low Cl⁻ concentration around Gawler River and Little Para River are wider in these aquifers than in the Quaternary aquifers, which is

consistent with the above observation that the head contours display a wider curvature around these rivers. c and d show analogous results for the CAP focus area. The Cl⁻ concentration is generally  lower than in the NAP focus area, especially in the mountain, most likely a result of lower evapotranspiration associated with the higher elevation of this area. A strong correlation between zones of low Cl⁻ concentration and stream locations can be seen in the basin. These zones appear wider and somewhat less distinct in the Tertiary aquifers than in the Quaternary aquifers, but it should be noted that the data density is lower in these aquifers. In both cases, a sharp change in Cl⁻ concentration can be seen at the transition between the mountain and the basin, therefore, suggesting that MBR is insignificant.~~this water originates from stream leakage in the basin (i.e. MFR). The possibility that this water originates from the higher elevation areas of the mountain – where salinity is low – through deep groundwater flowpaths is unlikely since the hydraulic head data analysis suggests a predominance of local flow systems in the Mount Lofty Ranges (section 3.2.1), and since groundwater salinity in the deep layers of the basin generally show high salinity. Furthermore, the Cl concentrations in the in-between streams zones (i.e. away from streams) in the basin aquifers are in most cases much higher than in the 'triangular facets' of the base of the mountain, and so this suggests that these 'triangular facets' do not contribute either to the basin recharge (or at least not in significant proportion).~~

The Cl⁻ concentration in streams running down from the Mount Lofty Ranges into the AP basin was analysed to investigate if stream leakage can explain the  groundwater Cl⁻ concentrations measured in the basin. A summary of available flow rate, electrical conductivity and derived Cl⁻ concentration data for six monitoring stations located near the transition between the mountain and the basin is presented in Table 1. The location of the stream gauges is indicated in Figure 2. The relationship between streamflow rate  and Cl⁻ concentration is shown from a scatter plot in Figure 12. The stream Cl⁻ concentration displays significant  variations, with a  decreasing trend as flow increases. The relationship between flow rate and Cl⁻ concentration nevertheless varies between different streams. This probably reflects,  different catchment characteristics that  are likely to influence the streamflow generation mechanisms (i.e., topography, climate, geology, landuse). The variations of streamflow rate and Cl⁻ concentration show a strong seasonality, as illustrated through selected time.  series  for  Gawler River and Brownhill Creek in Figure 13a and Figure 13b, respectively. These time series confirm that low Cl⁻ concentrations occur during  periods of high flow,  which coincide with the wet season (May–September). During this season, lower Cl⁻ concentrations in stream water can be explained by a relatively large contribution of overland flow or interflow, to streamflow generation, as these processes should experience little evapotranspiration relative to subsurface flow contribution. The significance of overland flow or interflow  is also supported by other observations mentioned above. During high flow periods, the infiltration potential in the mountain front zone should be enhanced due to higher stream water levels and wider wetted areas. We therefore  propose to estimate

concentration in streams as a representative value of thefor MFR Cl⁻ concentration by calculating the flow-weighted average Cl⁻ concentration in stream water. A more rigorous approach would require knowledge of the timing and rate of stream leakage, which are not available.. The flow-weighted average Cl⁻ concentration is 221, 115, 146, 67, 107 and 91 mg L⁻¹ in the North Para River, South Para River, Gawler River, Dry Creek, First Creek and Brownhill Creek, respectively (Table 1).

5  These data show that streams are a plausible source for the low Cl⁻ concentrations observed in the basin aquifers ((see Figure 10).

**4 Discussion**

**4.1 Strengths and limitations of using hydraulic head and chloride data**

One of the main strengths of hydraulic head data is that they can unambiguously indicate the contemporary flow direction (if

10  hydraulic conductivity can beis considered to be isotropic). In this study, the analysis of head contours gave indications that groundwater flows for a large part in local systems feeding streams in the Mount Lofty Ranges. It also allowed for the identification of losing stream conditions in the mountain front zone, where leakage from streams appears to recharge not only the Quaternary aquifers but also the deeper Tertiary aquifers. of the AP basin in significant proportions. Studying the head variation with depth in the basin gave further gave evidence that downward groundwater flowsflow from the

15  Quaternary aquifers to the underlying Tertiary aquifers, even though the is occurring. The rate of this flow is however unknown as long as the effective vertical hydraulic conductivity is unknown.

The main limitation of hydraulic head data is probably that they are quite sensitive to pumping., as illustrated through the present case study. This is problematic when the objective is to study the natural (i.e., pre-development) recharge mechanisms. Pumping in the AP basin mostly affects groundwater levels in the western part of basin, where large cones of

20  depression exist in the Tertiary aquifers due to extensive historical and ongoing pumping (Bresciani et al., 2015a).. Therefore, for the purpose of this study which focuses on the eastern part of the basin (where the mountain front zone is located), the issue may not be as critical. However, smaller-scale pumping wells surely also are likely to exist in the mountain front zone and in the mountain,as well and may affect the results to an unknown degree. ThisThe fact that the degree to which the hydraulic head data may have been distorted by pumping is unknown represents a non-negligible source

25  of uncertainty likely to be non-negligible.

Cl⁻ is a potentially an effectivegreat tool to distinguish between MFR and MBR. But for this, the stream water Cl⁻, but its usefulness depends on how different the sources are in terms of Cl concentration in . That is, if all the mountain front zone (i.e., potential MFR source) needs to be significantly different fromsources of recharge had the groundwatersame Cl⁻ concentration at, nothing could be learnt about recharge and flow mechanisms on the basebasis of the mountain (i.e.,

30  potential MBR source). DifferentCl data. Fortunately, different processes involved in the generation of MFR and MBR imply that these two potential sources of water may indeed are likely to have different Cl⁻ concentrations signatures (see section 2.2). This is certainly the case in the present case study, where the Cl⁻ concentration at the base of the mountain (i.e.

 is seen to be significantly different from that of the basin. In contrast, the Cl⁻ concentration in streams appears  to be similar to that of the low Cl⁻ concentration zones of the basin, which are aligned with  surface water features. These observations leave little doubt regarding  the recharge mechanisms in the AP basin, i.e., MFR appears to be the dominant recharge mechanism.

As for hydraulic heads, pumping can potentially distort the Cl⁻ concentrations from those of the natural system. In addition, irrigation may have influenced the Cl⁻ concentration in shallow wells located in agricultural areas. However, changes in solute concentrations are expected to be observed later than hydraulic head changes, and a dramatic shift in Cl⁻ concentrations is unlikely to be seen in Cl⁻ concentrations away from the main areas of pumping . Furthermore, if recharge from streams in the basin was only  a result of recent pumping, groundwater should have a very modern (i.e., post-development) recharge signature. Groundwater dating shows that this is not the case (Batlle-Aguilar et al., 2017). It can also be noted that the correlation between the zones of low groundwater salinity  and stream locations was already observed in the 1950s, i.e., using measurements anterior to the main  development period (Miles, 1952).

The interpretation of Cl⁻ data also relies on the assumptions of constant Cl⁻ inputs. This assumption is widely accepted in the literature for hydrogeological studies and in applications of the chloride mass balance method to estimate recharge (Wood, 1999; Scanlon et al., 2006; Crosbie et al., 2010; Healy and Scanlon, 2010). This assumption may not be strictly satisfied over the  AP basin because groundwater in the Tertiary aquifers can be quite old, as revealed by numerous samples showing paleo-meteoric origin (> 12,000 y) according to carbon-14 dating and noble gas measurements (Batlle-Aguilar et al., 2017). This indicates that different climatic conditions may have prevailed at the time of recharge, implying possible variations in Cl⁻ inputs. However, such old groundwater is mostly observed in the western part of the basin. Groundwater  in the mountain front zone is  younger (in most cases  < 10,000 y according to carbon-14 dating), making it less likely for these to reflect drastically different paleo-climatic conditions. Furthermore, even if the Cl⁻ inputs did vary over time, such temporal variations would not in itself explain the correlation of groundwater Cl⁻ concentration with stream locations in the basin.

Finally, the assumption of conservative Cl⁻ is deemed reasonable in view of the geology of the study area, which is not known to bear evaporite deposits such as halite. This assumption was also tested by analysing the chloride/bromide (Cl⁻/Br⁻) ratio from 161 well samples distributed over the study area. The average Cl⁻/Br⁻ ratio from these samples is 739 +/- 173 (molar) (Figure S13). This shows that groundwater has a similar Cl⁻/Br⁻ ratio to seawater (~650 molar), hence indicating that there is little water-rock interactions or dissolution of evaporite deposits (Drever, 1997; Davis et al., 1998). The low variance of the Cl⁻/Br⁻ ratio also suggests that the dominant process for the increasing salinity in the groundwater is evapotranspiration (Cartwright et al., 2006). Furthermore, even if Cl⁻ was not strictly conservative, focused recharge from streams in the basin

would again appear necessary to explain the fact that the zones of low groundwater Cl⁻ concentration are found along streams.

Compared to other methods that use noble gas, radioactive or isotopic tracers, the above approach appears simpler, more cost effective, and more reliable due to the much higher data density generally achievable (i.e.,~~Finally, the assumption of conservative Cl is deemed reasonable because chloride is usually not strongly adsorbed to mineral surfaces, and the aquifer materials in the study area are not expected to be a significant source of Cl in comparison to atmospheric inputs. In addition, even if Cl was not strictly conservative, leakage from streams would again appear necessary to explain the observed correlation of groundwater Cl concentration with streams in the basin.~~

given current technologies and budget constraints). Nevertheless, in contrast with some other methods (e.g., noble gases), it should be noted that this approach is only qualitative. I.e., it does not allow for the quantification of the relative proportion of MFR and MBR and their absolute rate. One way to extend the ideas presented in this study to gain more quantitative insight would be to use the data as calibration targets in a groundwater flow and Cl⁻ transport model. This is also the subject of ongoing efforts (Bresciani et al., 2015b).

**4.2 MFR versus MBR in the AP basin**

In an early study, Miles (1952) noted that the pre-development groundwater levels along the mountain front of the AP basin were reflective of unconfined conditions, and that the subsurface materials in this zone were favourable to stream infiltration. In addition, Miles (1952) already analysed the groundwater salinity distribution. He  observed that salinity contours were forming fan-shaped zones of low salinity "mushrooming" outwards from streams, with such patterns being visible up to more than 100 m below the ground surface. He concluded that stream infiltration along the mountain front zone was a major recharge mechanism for the basin aquifers. Later, in a study of the NAP aquifers, Shepherd (1975) arrived to the same conclusion, partly using similar arguments and further noting that: (i) groundwater hydrographs in the Quaternary aquifers were each year showing a rapid rise in water level shortly after Gawler River and Little Para River started to flow; (ii) the vertical head gradient and vertical hydraulic conductivity were indicative of significant downward flow from the Quaternary to the Tertiary aquifers. Additionally, a number of studies directly measured groundwater gains and losses using differential flow-gauging along streams entering the AP basin (Hutton, 1977; Green et al., 2010; Cranswick and Cook, 2015). All, found that several streams were losing a significant amount of water in the mountain front zone. Finally, Zulfic et al. (2010) found that bore yield based on air-lift testing conducted at the time of drilling (e.g., Williams et al., 2004) in the Mount Lofty Ranges did not increase beyond 100 m depth for most geology types. This finding can be interpreted as hydraulic conductivity being relatively low beyond that depth. This would promote  local groundwater flow systems in the mountain, in line with the current analysis.

In contrast, Gerges (1999) and Batlle-Aguilar et al. (2017) proposed that MBR is the dominant recharge mechanism for the Tertiary aquifers of the AP basin. A major argument  in these studies was based on the observation that salinity is generally higher in the Quaternary aquifers than in the T1 and T2 aquifers. From this observation, the authors suggested that the relatively fresh water found in the T1 and T2 aquifers could not be the result of downward leakage. Instead, they proposed,  that this water should come from the Mount Lofty Ranges (where the salinity is lower) through subsurface flow. However, along streams, the Cl⁻ concentration in the Quaternary aquifers is in fact very similar to that of the underlying Tertiary aquifers ( (Figure 10). Furthermore, the possibility that this relatively fresh water originates from the higher elevation areas of the Mount Lofty Ranges through deep groundwater flow paths is unlikely since: (i) the hydraulic head data suggest a predominance of local flow systems in the Mount Lofty Ranges (section 3.2.1); (ii) there is an important mismatch between groundwater Cl⁻ concentrations at the base of the mountain block and those in the upper part of the basin (section 3.2.2); and (iii) groundwater in the deep layers of the basin (i.e., T3 and T4 aquifers) generally shows higher salinity (Gerges, 1999), implying that deep groundwater flow paths cannot explain the observed fresh water. Another argument in).  Batlle-Aguilar et al. (2017) was based on the observation that relatively old groundwater was measured near the top of Tertiary aquifers (from carbon-14 dating). From this observation, the authors suggested that groundwater could not be recharged in the basin, but rather further away, in the Mount Lofty Ranges. However, most of the old groundwater samples analysed were from wells found quite some distance away from the mountain front zone, where the Tertiary aquifers become confined –logically implying an increase of age with distance from the recharge zone.. Furthermore,  relatively young groundwater was found at significant depth near major faults of the basin, precisely suggesting the occurrence of focused recharge (Batlle-Aguilar et al., 2017). Finally, neither Gerges (1999) nor Batlle-Aguilar et al. (2017) proposed a mechanism to explain how the groundwater could be more saline outside of the low-salinity corridors under MBR-prevailing conditions, as  consistently observed  in both the Quaternary aquifers and the Tertiary aquifers (Figure 10). These zones of higher salinity directly contradict the MBR hypothesis, including MBR that  would be derived from the recharge over triangular facets at the base of the mountain. It seems more likely that the groundwater of higher salinity in the basin originates from  diffuse recharge, which  would naturally imply a much higher salinity as a result of evapotranspiration than focused recharge from  streams .

Hence, on the basis of  robust consistent evidence given in this work (including consistent findings from hydraulic head and Cl⁻ analyses) and through a critical review of earlier investigations, we propose that MFR is the most plausible and predominant recharge mechanism for the relatively fresh water found in the  AP basin (i.e., as in Figure 1b). This finding is expected to have important consequences for

future investigations and for the management of water resources in the region of Adelaide region. A conceptual model depicting the suggested recharge mechanisms, and how thesethey can explain the observed Cl⁻ (or salinity) patterns – at least in the eastern part of the basin –, is shown in Figure 1413.

**5 Conclusion**

WeThis study presented and demonstrated through a regional-scale an example the use effectiveness of using hydraulic head, Cl⁻ and EC data to distinguish between MFR and MBR to basin aquifers. Hydraulic heads can inform on groundwater flow directions and stream-aquifer interactions, while chloride concentrations and EC values can be used to distinguish between different water sources if these have a distinct signature. This information can provide evidence for the occurrence or absence of MFR and MBR.

In the above case study, bothThe most useful way of using hydraulic head and Cl⁻ (equivalently, EC) analyses gave informative and consistent results (i.e., both suggested a predominance of MFR), which gives confidence in the interpretation. The Cl⁻ data is through the analysis was particularly straightforward and authoritative, and it further allowed for the identification of diffuse recharge in the basin.

Difficulties in the interpretationshape of hydraulic head and Cl⁻ data may arise for particularcontours adjacent to surface water features to identify losing and gaining stream conditions such in the mountain front zoneas when pumping effects are significant. However, the study well as in the mountain itself. Other useful ways of the AP basin demonstrates that even for a basin that has been subject to long-term groundwater extraction (for about a century in this case), the data can allow for the identification of natural recharge mechanisms. Cl⁻ in particular is expected to be more robust than using hydraulic head to pumping effects.

The relevance of the data were presented approach lies in that the variables used (hydraulic heads, Cl⁻ concentrations and EC values) have been routinely measured for decades in many parts: analysis of the degree of correlation of the world; if not, their measurement is inexpensive provided wells already exist. While these data cannot tell us everything (e.g., they cannot directly inform on the recharge rates), we expect that in many cases a significant dataset would be readily available, bearing an as-yet unexploited potential to inform on the recharge mechanisms. Such information is critical for conceptualizing and managing groundwater systems.

The AP basin serves as an examplehydraulic head with topography, comparison of a region where the recharge mechanisms have long been debated in the contextstream levels with nearby groundwater levels, and evaluation of groundwater resources management. Applicationthe vertical head gradient in the mountain front zone. However, the latter three methods can only indicate the flow direction, while the significance of the proposed rationale revealed to be effective in resolving this debate.

It is expected that the findings of this study will have significant impacts on the management of water resources in the Adelaide regionthis 
[revised manuscript text omitted]

**(a) Physical configuration**

*Basin floor* | *Mountain front* | *Mountain block*

Mountain

Fault

Basin

**(b) MFR**

Groundwater flowpath

Surface water flowpath

**(c) MBR**

Groundwater flowpath

Surface water flowpath

**(d) MFR and MBR**

Groundwater flowpath

Surface water flowpath

(a) Physical configuration

Basin floor | Mountain front | Mountain block

Mountain

Fault

Basin

(b) MFR dominates

(c) MBR dominates

(d) MFR and MBR are significant

**Figure 1. Conceptual models of the transition between mountain and basin:. (a) physicalPhysical configuration and. (b-c) Three possible conceptualizations of regarding the MSR whereto the basin: (b) MFR dominates, (c) MBR dominates, and (d) both MFR and MBR are significant.**

[Figure]

[Figure]

**Figure 2.** Situation map showing elevation (in mAHD, i.e., Australian Height Datum) and relevant features of this study.

[Figure]

**Figure 3.** Schematic diagram showing triangular facets at the base of the mountain block (after Welch and Allen (2012)).

[Figure]

**Figure 4. Schematic diagram showing hydraulic head contours and groundwater flow directions in the horizontal plane for (a) losing and (b) gaining stream conditions (after Winter et al. (1998)).**

[Figure]

**Figure 5.** Impact of a difference in basement elevation induced as a result of faulting on hydraulic head in a hypothetical setting. (a) Hydraulic conductivity field in a vertical cross-section representing a sedimentary layer  overlying a basement having significantly lower hydraulic conductivity with the fault zone  itself having no different hydraulic conductivity (i.e. the fault only implies the difference in basement elevation). (b) Results from an unconfined groundwater flow simulation in which a constant head (80 m) was specified on the left boundary and inflow was specified on the right boundary (at a rate proportional to the hydraulic conductivity). A sharp difference in hydraulic head is observed across the fault zone. The simulation was performed using MODFLOW-NWT (Niswonger et al., 2011), with 200 cells in horizontal direction and 150 cells in the vertical direction.

(Niswonger et al., 2011) with uniform grid spacing (200 cells in horizontal direction × 150 cells in the vertical direction),

[Figure]

**Figure 3. AP basin and AP catchment based on surface topography. Elevations are in mAHD (Australian Height Datum, i.e. above mean sea level).**

[Figure]

Formatted Table

[Figure]

**Figure 6.** Total thickness of the Quaternary (Q) sediments (a) and Tertiary (T) sediments (b) in the AP basin. The colour schemes in (a) and (b) are different.~~scheme is different for the two figures. The maps were constructed by calculating the difference between the land surface elevation and the bottom elevation of the Quaternary sediments (a), and between the bottom elevation of the Quaternary sediments and the top elevation of the basement (b) (elevation surfaces by courtesy of the Department of Environment, Water and Natural Resources, South Australia).~~

(a)

[Figure]

(b)

[Figure]

[Figure]

**Figure 7**. Schematic hydrogeological cross-sections in (a) the NAP and (b) the CAP sub-basins (see Figure 2) (after Shepherd (1975); Gerges (1999); Zulfic et al. (2008); Baird (2010)). Q: Quaternary aquifers (lumped together; in reality there are up to six different aquifers depending on location, separated by clay layers); T1, T2, T3 and T4: Tertiary aquifers; UTS: Undifferentiated Tertiary Sand aquifers; MPC: Munno Para Clay aquitard; BPF: Blanche Point Formation aquitard; FR: Fractured-Rock aquifers; RF: Redbank Fault; AF: Alma Fault; PF: Para Fault; HVF: Hope Valley Fault; EBF: Eden-Burnside Fault.

[Figure]

Figure 8. Cl⁻ versus EC data in the AP catchment. The fitted function used to describe the relationship is $[Cl^-] = [Cl] = 0.04411 \times [EC]^{1.208}$, where [Cl⁻] and [EC] are in units of mg L⁻¹ and µS cm⁻¹, respectively (coefficient of determination: $R^2 = 0.9996$).

[Figure]

| | | | | | |
|---|---|---|---|---|---|
| ······ Topographic Contour (mAHD) | | · Hydraulic Head Data Point | —— Stream (order > 2) | 🟥 NAP Boundary | 🟧 Mountain |
| —— Head Contour (mAHD) | | ⇢ Groundwater Flow Direction | —— Fault | 🟩 CAP Boundary | 🟩 Basin |

[Figure]

**Figure 9. Head and topographic contours in the NAP (a-b) and CAP (c-d) focus areas. On the basin side, (a) and (c) show the head in the Quaternary aquifers, while (b) and (d) show the head in the Tertiary aquifers. On the mountain side, all four sub-figures show the head in the and Mount Lofty Ranges aquifers. In (a-b), the contour interval is 10 m in the basin and 40 m in the mountain (both for head and topography), while in (c-d), it is 20 m in the basin and 80 m in the mountain. Selected groundwater flow directions highlight apparent gaining stream conditions in the lower part of the mountain and loosing stream conditions in the mountain front zone.**

[Figure]

values, indicating a potential for upward flow (i.e., groundwater discharge). in the mountain. Topographic contours are also indicated. The contour interval is different in the basin (10 m) and in the Mount Lofty Ranges (40 m) both for head and topographic contours to accommodate the fact that the range of variations is much larger in the mountain than in the basin. Values are in mAHD (Australian Height Datum).

[Figure]

[Figure]

Figure 7. Head contoursFigure 11. Cl⁻ concentration in the NAP (a-b) and CAP (c-d) focus areas. On the basin side, (a) and (c) show the Cl⁻ concentration in the Quaternary aquifers, while (b) and (d) show the Cl⁻ concentration in the Tertiary aquifers. On the mountain side, all four sub-figures show the Cl⁻ concentration in the and Mount Lofty Ranges aquifers (a), and in the Tertiary aquifers and Mount Lofty Ranges aquifers (b). Greenish colours are associated with head values in the basin, while brownish colours are associated with head values in the mountain. The head contour interval is different in the basin (10 m) and in the Mount Lofty Ranges (40 m) to accommodate the fact that the range of variations is much larger in the mountain than in the basin. Values are in mAHD (Australian Height Datum).

[Figure]

**Figure 8.** Result of the subtraction, at every point, of the nearest river elevation by the hydraulic head, in the Quaternary aquifers and Mount Lofty Ranges aquifers (a), and in the Tertiary aquifers and Mount Lofty Ranges aquifers (b). The same colour scheme is applied everywhere. Blueish colours are for negative values, indicating that the nearest river level is lower than the hydraulic head, while reddish colours are for positive values, indicating that the nearest river level is higher than the hydraulic head.

[Figure]

**Figure 9. Head difference between the Quaternary aquifers (Q) and the Tertiary aquifers (T). Blueish colours are for negative values, indicating that the head is lower in Q than in T, while reddish colours are for positive values, indicating that the head is higher in Q than in T.**

[Figure]

Figure 10. and is indicated in (a). The latterChloride concentration in the Quaternary aquifers and Mount Lofty Ranges aquifers (a), and in the Tertiary aquifers and Mount Lofty Ranges aquifers (b). The colour scheme is chosen favourable to the study of relatively low salinity zones. I, i.e., all Cl⁻ concentrations larger than , [Cl] < 1,400 mg L⁻¹ are included in the same class (red colour), but in reality(or [EC] < 5,327 µS cm⁻¹) for the purpose of this study; much higher values exist, and are all represented in red.

[Figure]

[Figure]

**Figure 11. Figure 12.** Scatter plot of chloride concentration versus streamflow ratestream flow for six streams running down from the Mount Lofty Ranges into the AP basin.

[Figure]

[Figure]

Figure 12. Surface Figure 13. Stream water chloride concentration and flow rate over selected periodsdata for (a) Gawler River (a) and (b) Brownhill Creek. The (b). Notice the different scales on the axes in (a) and (b) have a different scale.

[Figure]

[Figure]

Figure 14.

Figure 13. Conceptual model of the recharge mechanisms for the aquifers of the AP basin aquifers, as seen in a cross-section perpendicular to (and centred on) a stream in the mountain front zone, and how thesethey can explain the observed salinity patterns in the eastern part of the basin. In blue: groundwater salinity. Blue to red colours indicatehaving relatively low tosalinity; in red: groundwater having relatively high salinity.